# Complete microtubule–kinetochore occupancy favours the segregation of merotelic attachments

Damian Dudka[1], Anna Noatynska[1], Chris A. Smith[2,5], Nicolas Liaudet[3], Andrew D. McAinsh[2] & Patrick Meraldi[1,4]

Kinetochores are multi-protein complexes that power chromosome movements by tracking microtubules plus-ends in the mitotic spindle. Human kinetochores bind up to 20 microtubules, even though single microtubules can generate sufficient force to move chromosomes. Here, we show that high microtubule occupancy at kinetochores ensures robust chromosome segregation by providing a strong mechanical force that favours segregation of merotelic attachments during anaphase. Using low doses of the microtubules-targeting agent BAL27862 we reduce microtubule occupancy and observe that spindle morphology is unaffected and bi-oriented kinetochores can still oscillate with normal intra-kinetochore distances. Inter-kinetochore stretching is, however, dramatically reduced. The reduction in microtubule occupancy and inter-kinetochore stretching does not delay satisfaction of the spindle assembly checkpoint or induce microtubule detachment via Aurora-B kinase, which was so far thought to release microtubules from kinetochores under low stretching. Rather, partial microtubule occupancy slows down anaphase A and increases incidences of lagging chromosomes due to merotelically attached kinetochores.

[1] Department of Cell Physiology and Metabolism, Faculty of Medicine, University of Geneva, 1211 Geneva 4, Switzerland. [2] Centre for Mechanochemical Cell Biology & Division of Biomedical Sciences, Warwick Medical School, University of Warwick, CV4 7AL Coventry, UK. [3] Bioimaging Facility, Faculty of Medicine, University of Geneva, 1211 Geneva 4, Switzerland. [4] Translational Research Centre in Onco-hematology, Faculty of Medicine, University of Geneva, 1211 Geneva 4, Switzerland. [5] Present address: Metabolic Research Laboratories and MRC Metabolic Diseases Unit, Institute of Metabolic Science, Addenbrooke's Hospital, Hills Road, Cambridge CB2 0QQ, UK. These authors contributed equally: Damian Dudka, Anna Noatynska. Correspondence and requests for materials should be addressed to P.M. (email: patrick.meraldi@unige.ch)

Mitotic cells assemble a bipolar mitotic spindle formed by microtubules (MTs) emanating from the spindle poles. MTs "search and capture" chromosomes via kinetochores (KTs), protein complexes assembled on the centromeric DNA[1]. KTs utilize the forces generated by MT assembly/disassembly to drive chromosome movements. Before anaphase, all sister-KT pairs must form bi-oriented attachments and align at the spindle equator. While budding yeast KTs only bind a single MT[2, 3], vertebrate KTs bind multiple MTs (15–20 MTs in human cells)[4]. Although a single depolymerizing MT can generate up to 30 pN of force[5], as little as 0.1 pN is enough to move a vertebrate chromosome in the cytoplasm[6–8], raising the question as to why human KTs evolved to accommodate 20 MTs.

One explanation is that multiple MTs are required to stretch the sister-KTs apart: depolymerizing KT–MTs pull sister-KTs towards opposite spindle poles, increasing the inter-KT distances and stretching the centromeric chromatin. This stretching in turn generates an opposing spring force (tension), which pulls on KT–MTs. Micromanipulation experiments in metazoan cells[9,10] or biophysical measurements with purified yeast KTs[11,12] demonstrated how tension stabilizes KT–MT attachments. MT occupancy and the average inter-KT distance increase as cells progress from prometaphase to metaphase, implying that MT occupancy and tension mutually reinforce each other[11]. Moreover, MT occupancy and inter-KT stretching have been linked to the satisfaction of the spindle assembly checkpoint (SAC) and the correction of erroneous KT–MT attachments[13].

The SAC prevents chromosome segregation errors, by delaying anaphase onset until the last KT forms end-on MT attachments[14]. KTs lacking end-on attachments recruit the SAC kinase Mps1 (monopolar spindle 1), initiating a signalling cascade that recruits and activates the checkpoint proteins Mad2 (mitotic arrest deficient 2) and BubR1 (budding uninhibited by benzimidazole-related 1), and ultimately blocks anaphase onset and sister chromatid separation. Since Mps1 and MTs bind the Ndc80 complex, the main MT-binding site at KTs, in a mutually exclusive manner[15,16], MT attachment removes all checkpoint proteins from KTs and satisfies the SAC. In theory, a checkpoint that can only be satisfied once a complete set of MTs binds all KTs would give rise to an ultra-sensitive checkpoint response, thus ensuring a robust attachment at anaphase onset. It remains, however, unclear exactly how many MTs must bind to a KT to satisfy the SAC: one study found that the SAC protein Mad1 (mitotic arrest deficient 1) starts to detach from KTs at 50% MT occupancy[17], while another study found that unaligned bi-orientated KTs with an incomplete set of KT–MT attachments still had high levels of the SAC protein Bub1 (budding uninhibited by benzimidazole 1)[18].

Inter-KT stretching has also been long discussed as a prerequisite for SAC satisfaction, since it reflects bi-orientation[19]. Whether this is the case is difficult to address in human cells, since the most frequently used tool to lower inter-KT stretching, the MT-stabilizing drug taxol, also leads to unattached KTs[20]. Nevertheless, studies using human cells with unreplicated chromosomes, or which express Ndc80 mutants that over-stabilize KT–MT attachments, showed that after an initial delay, the SAC is satisfied despite minimal or no inter-KT stretching, demonstrating that it is not an absolute requirement[21–23]. This is consistent with a study in *Drosophila* cells that found no correlation between inter-KT distances and SAC satisfaction[24] and a study reporting that human cells can mount a SAC response despite normal inter-KT distances[25]. Nevertheless, inter-KT stretching is still thought to be the important criterion that cells use to distinguish between bi-oriented and erroneous syntelic KT–MT attachments. In syntelic attachments both sister-KTs are bound by MTs oriented towards the same spindle pole, resulting in low inter-KT distances. These attachments are corrected by the kinase Aurora-B[13,26]. According to the current model, Aurora-B activity is spatially restricted to centromeric chromatin, between the sister-KTs. When inter-KT distances are low, Aurora-B phosphorylates the Ndc80 complex, promoting KT–MT detachment; when inter-KT distances are high, Aurora-B cannot reach its substrates, leaving attachments intact. This is supported by the fact that Aurora-B phosphorylates the Mis12 and Ncd80 complexes in a proximity-dependent manner[27,28], and that forced localization of Aurora-B onto KTs induces MT detachment[29]. The Aurora-B gradient model thus predicts that sister-KTs with low inter-KT distances should detach their chromosomes, irrespective of the type of attachment. It has been, however, challenged in budding yeast, as cells with ubiquitously located Ipl1/Aurora-B still correct syntelic KT–MT attachments[30].

Here, we addressed the role of MT occupancy and inter-KT stretching in human cells using BAL27862, a new MT-targeting agent that blocks MT growth with distinct effects on MT organization[31], activating the SAC at optimal anti-proliferative concentrations[32]. The prodrug of BAL27862, BAL101553, is currently being evaluated for the treatment of advanced cancer patients in phase 1/2A clinical trials. We find that at the low nanomolar, sub-cytotoxic range BAL27862 reduces MT occupancy and impairs inter-KT stretching, but does not lead to unattached KTs, unlike low doses of other classical MT-targeting agents, such as nocodazole, noscapine, or vinblastine[33–35]. Using this tool we demonstrate that a reduced MT occupancy and an impaired inter-KT stretching does neither prevent SAC nor activate Aurora-B-dependent MT detachment. Instead, we find that full MT occupancy is required for robust anaphase A forces to ensure the segregation of lagging, merotelic chromosomes.

## Results

**BAL27862 reduces KT–MT occupancy.** To investigate why human KTs need to bind a high number of MTs, we applied a novel MT-targeting agent, BAL27862[31], to reduce their number. To test whether BAL27862 affects KT–MT occupancy we used transmission electron microscopy (EM), the gold standard for the quantification of MTs in KT fibres[33,36–38]. Non-cancerous human retina pigment epithelial cells immortalized with telomerase (hTert-RPE1) that express the KT marker GFP-CENPA (green fluorescent protein-centromere protein A) were treated for 4 h with a sub-cytotoxic dose (12 nM) of BAL27862. Since KTs in metaphase cells have the highest MT occupancy[39], we synchronized cells in metaphase with the proteasome inhibitor MG132 for 2 h, and examined orthogonal sections in which chromosomes formed a metaphase plate (Fig. 1a). Based on criteria used in previous studies (see Methods section[37,38]), we found that KT fibres in BAL27862-treated cells contained a median of 13.5 (11–16; 95% confidence interval (CI); N = 3) MTs versus 19.8 (17.5–21; 95% CI; N = 3) MTs in dimethyl sulfoxide (DMSO)-treated cells (Fig. 1a, b). To corroborate these results we also measured the fluorescence intensity of KT-MTs in hTert-RPE1 cells expressing eGFP-α-tubulin. MG132-arrested cells were fixed, stained with CENPA antibodies to mark KTs[40], and imaged by fluorescence microscopy: k-fibres were identified based on the position of the CENPA signal and the intensity of their KT-proximal part (400 nm from the CENPA centroid signal) quantified with a custom written MATLAB-based code (>1000 KT fibres per condition). We found a 35.5% decrease in KT fibre intensity (Fig. 1c, d), corroborating our EM results. Despite a lower number of MTs in the KT fibres, there were no evident spindle morphology or chromosome alignment defects in BAL27862-treated cells (Fig. 1e). Moreover, spindles were only

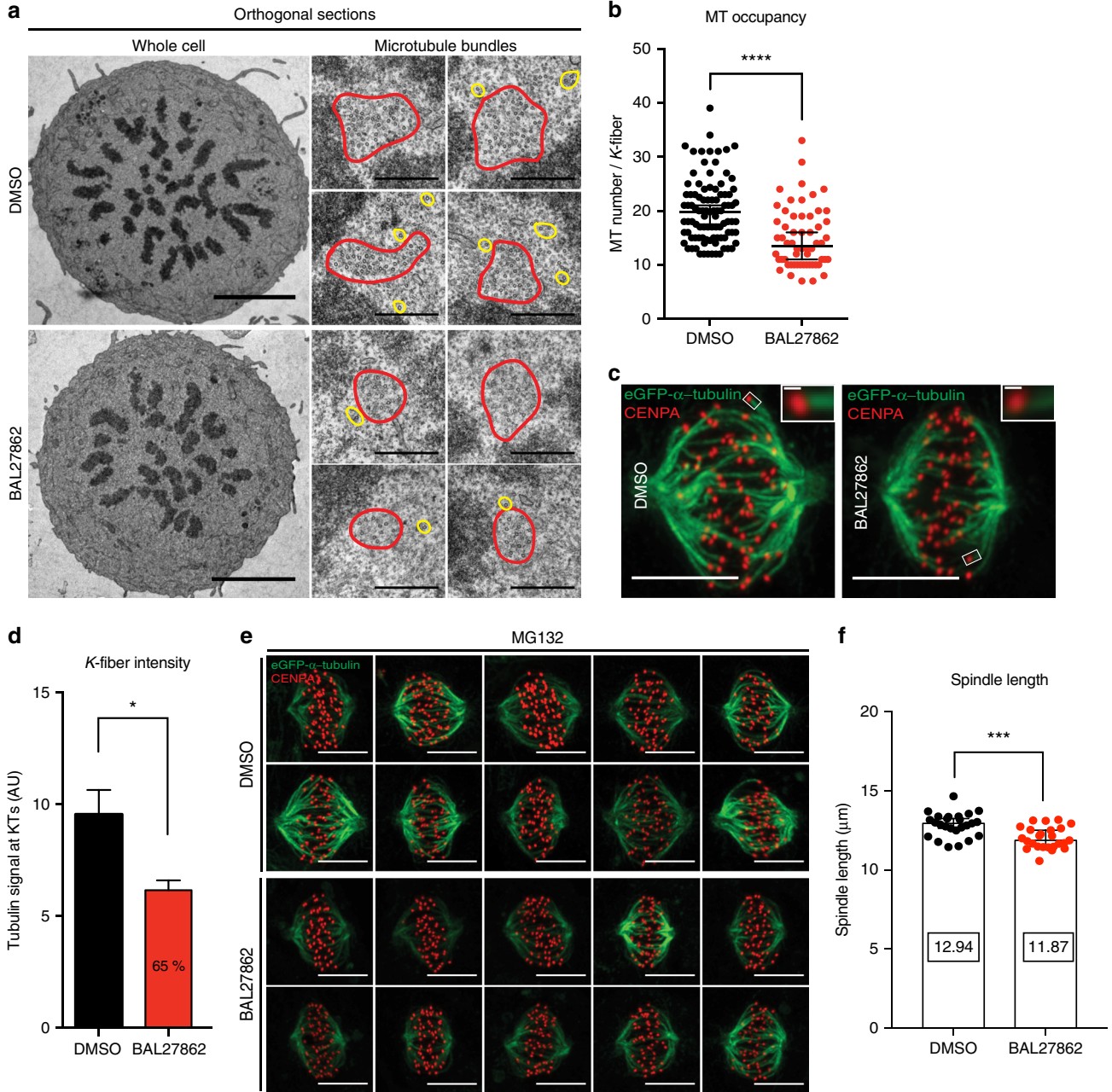

**Fig. 1** BAL27862 reduces KT–MT occupancy. **a** Transmission electron microscopy orthogonal sections of hTert-RPE1 GFP-CENPA cells treated with DMSO or 12 nM BAL27862 and blocked in metaphase with 10 μM MG132. Left panels show whole cells at low magnification. Scale bars = 5 μm. Right panel shows individual KT fibres (red circles) or inter-polar MTs (yellow circles). KT fibres were defined as a set of at least 7 parallel MTs, in the vicinity of chromatin and not more than 80 nm separated from one another[38]. Scale bars = 500 nm. **b** Quantification of the number of MTs in KT–MT fibres in hTert-RPE1 GFP-CENPA cells treated with DMSO or 12 nM BAL27862. Data are represented as a dot plot; error bars represent 95% CI of medians; each dot represents a single KT fibre ($N = 3$ independent experiments; $n = 59$–129 KT fibres, $p < 10^{-4}$, two-tailed Mann–Whitney $U$-test). **c** Immunofluorescence images of hTert-RPE1 eGFP-α-tubulin (green) cells treated either with DMSO or 12 nM of BAL27862 (4 h) and 10 μM of MG132 (2 h) and stained with anti-CENPA antibody (red). Scale bars = 5 μm. Insets show a KT-proximity area measurement. Scale bars = 250 nm. **d** Quantification of eGFP-α-tubulin intensities at KTs in DMSO and BAL27862-treated cells. $N = 3$, $n = 1135$–1156 $k$-fibres from 29 to 30 metaphase-arrested cells per condition, error bars represent s.e.m., $p = 0.045$, two-tailed $t$-test. **e** Immunofluorescence images of 10 randomly selected hTert-RPE1 eGFP-α-tubulin cells from a single experiment. Cells were treated in parallel with DMSO or 12 nM BAL27862, 10 μM of MG132, and stained with anti-CENPA antibodies (red). Scale bars = 5 μm. **f** Quantification of spindle length in hTert-RPE1 Centrin1-GFP/GFP-CENPA cells treated with DMSO or 12 nM BAL27862. $N = 4$, $n = 25$ cells; error bars represent 95% CI of medians, $p = 0.0002$, two-tailed $t$-test. *, **, ***, **** represent $p < 0.05$, $< 0.01$, $< 0.001$, $< 0.0001$ respectively

marginally shorter when measured from live cell images of hTert-RPE1 Centrin1-GFP (centrosome marker)/GFP-CENPA cells (Fig. 1f). We conclude that nanomolar concentrations of BAL27862 decrease MT occupancy at KTs by ~1/3rd, while preserving bipolar spindle integrity.

**MT occupancy controls inter-KT stretching.** To test whether reduced MT occupancy affects KT conformation, we investigated whether BAL27862 affected the distance between inner and outer KT structures. To do this we determined the sub-pixel position of GFP-CENPA (centromere/inner KT) and the amino-terminus of

Ndc80 (outer KT; using 9G3 antibody) using a MATLAB-based tracking assay (Fig. 2a,b)[41,42]. From the positions of GFP-CENPA and Ndc80 signals in sister-KT pairs we calculated, following correction of Euclidean distances[43], the intra-KT distance (~103 ± 2.6 nm in control, consistent with earlier studies;[44] $N = 4$) and the angle between the intra-KT axis and the sister-KT axis (called swivel or $k$-tilt; Fig. 2c–e)[20,42]. Previous measurements in HeLa cells found that this intra-KT distance is reduced when KTs are unattached (nocodazole) and that the swivel angle is increased[42]. Our measurements of BAL27862-treated hTert-RPE1 cells revealed a minimal increase in intra-KT distances (4.4 ± 2.4 nm s.

d.; $N = 4$) and slight reduction in swivel (7.5° decrease in s.d.; $N = 4$; Fig. 2d, e), suggesting that BAL27862-treated cells have the expected conformation for bi-oriented attachments.

To investigate how a reduction in MT occupancy affects KT movements, we used an automated four-dimensional (4D) KT-tracking assay to analyse the dynamics of GFP-CENPA in hTert-RPE1 cells during metaphase (Supplementary Movie 1 and 2). Bi-oriented attachment stretches the distance between sister-KTs and leads to semi-periodic oscillatory movements that can be revealed with an autocorrelation analysis of the KT positions (Fig. 2f):[45] the position of the first minimum indicates the average

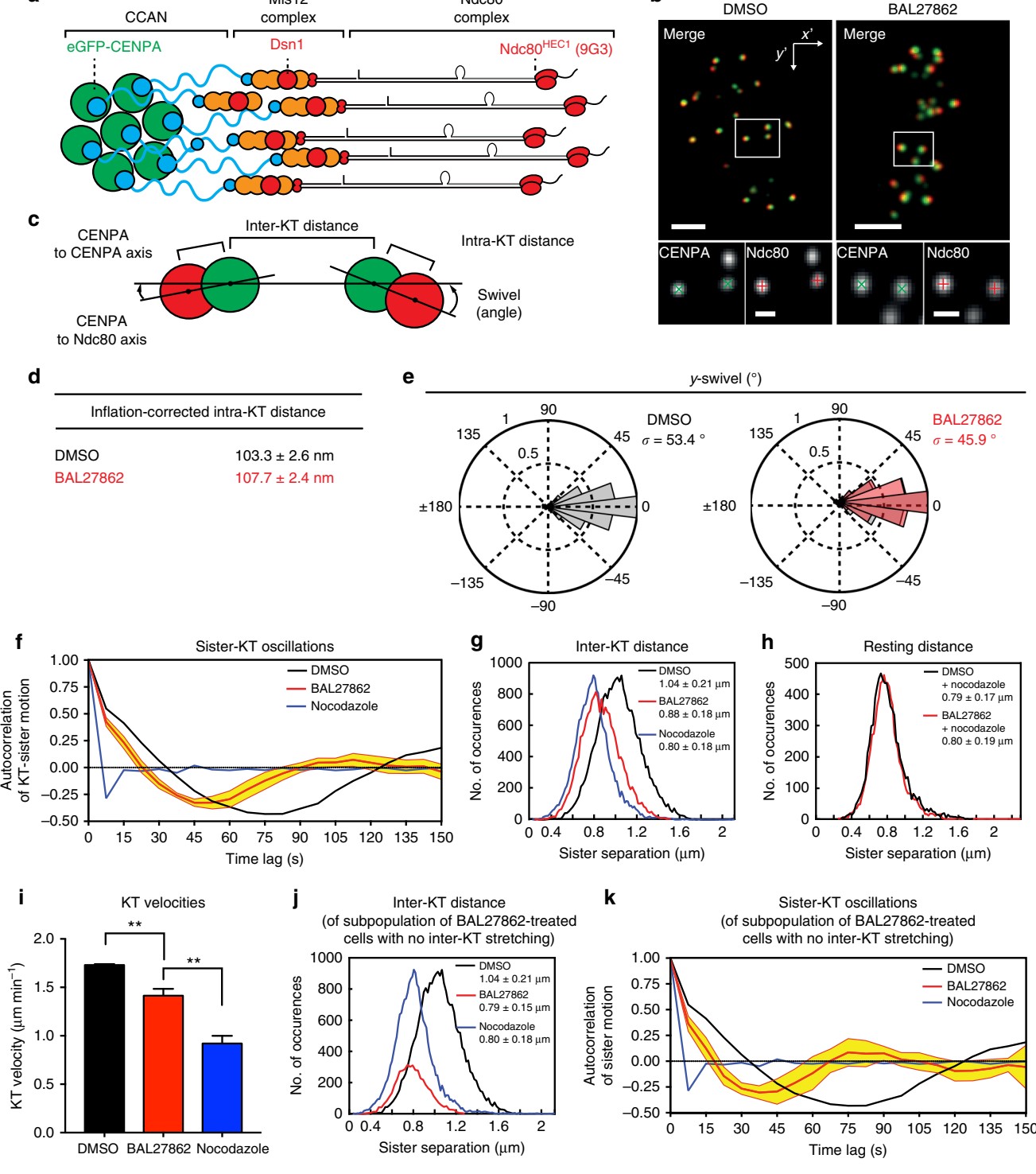

half-period, while the magnitude reflects the abundance of sister-pairs oscillating at that frequency. Both parameters depend on bi-oriented attachment, since high doses of nocodazole abolish oscillations and lower inter-KT distances (Fig. 2f, g). Treatment with 12 nM of BAL27862 dramatically reduced average inter-KT distances to a value that is only marginally larger than that measured in nocodazole-treated cells (Fig. 2g). This suggests a severe reduction in the forces acting on KTs. The reduction in inter-KT distances was not caused by changes in the resting length of the centromeric chromatin, as inter-KT distances in cells treated with high doses of nocodazole were the same whether BAL27862 was added or not (Fig. 2h). Nevertheless, BAL27862 treatment still allowed for oscillatory movements, albeit with a shortened half-period (45–60 s, vs. 75–90 s in DMSO-treated cells) and a reduced velocity ($1.41 \pm 0.07 \, \mu m \, min^{-1}$ s.e.m. vs. 1.73 $\mu m \, min^{-1} \pm 0.01 \, \mu m \, min^{-1}$ s.e.m. in DMSO-treated cells; $N = $ 3–4), indicating maintenance of bi-oriented attachments and dynamic KT-attached MTs (Fig. 2f, i). Importantly, ~1/4th of the BAL27862-treated cells had an average inter-KT distance similar to that measured in nocodazole-treated cells ($0.80 \pm 0.18 \, \mu m$ s.d.; $N = $ 2–4; Fig. 2j). This indicated a population of BAL27862-treated cells with no stretching/force generation. The sister-KTs in this subpopulation of cells, nevertheless, displayed semi-regular oscillations with a shortened period (Fig. 2k). Taken together, these data show how low MT occupancy at KTs dramatically reduces average inter-KT stretching, while still allowing for bi-oriented attachment and generation of oscillatory movements.

**Reduced MT occupancy does not prevent mitotic progression.**
The intact spindle morphology and lack of obvious chromosome alignment defects in BAL27862-treated cells (Fig. 1e) gave us the opportunity to test whether the combination of decreased MT occupancy and low inter-KT stretching allow for SAC satisfaction. hTert-RPE1 cells expressing H2B-mCherry (chromosome marker) and EB3-eGFP (MT marker) were treated with 12 nM BAL27862 and recorded every 3 min by time lapse imaging (Supplementary Movie 3 and 4). Cells initiated anaphase (median time of $18 \pm 3$ min 95% CI.; $N = 3$) with only a 3-min delay when compared to DMSO-treated cells (median time of 15 min $\pm$ 3 min; $N = 3$; Fig. 3a). This delay arose during chromosome alignment (median time of $9 \pm 3$ min vs. 6 min $\pm 0$ min in DMSO-treated cells; $N = 3$), as mitotic progression was normal after the final chromosome aligned (Fig. 3b, c). The absence of a longer delay cannot be attributed to effects of BAL27862 on the SAC itself, because higher doses of the drug (33 nM) led to a permanent mitotic arrest that was alleviated by the Mps1 inhibitor reversine (Fig. 3a).

Given the broad range of inter-KT distances (Fig. 2g), it was possible that BAL27862 would only delay the establishment of high inter-KT distance, which could explain how cells would satisfy the checkpoint given sufficient time. To test this we tracked single hTert-RPE1 GFP-CENPA cells treated with DMSO or 12 nM BAL27862 and assessed their inter-KT distances at anaphase onset (Fig. 3d–f). For comparison cells were also treated with high doses of nocodazole to determine the typical distribution of inter-KTs in the absence of MTs, which we found to be centred on 0.77 μm with 80% of all the values comprised between 0.70 and 0.87 μm ($N = 2$; Fig. 3f). DMSO-treated cells entered anaphase with a median inter-KT distance of 1.05 μm ($\pm 0.07$ μm 95% CI; $N = 4$); all 33 cells entering anaphase had average inter-KT distance higher than 0.93 μm, well above the resting distance distribution (Fig. 2h). The average inter-KT distance at anaphase in BAL27862-treated cells was lower ($0.91 \pm 0.13$ μm 95% CI; $N = 3$), and 9 out of 19 cells had an average inter-KT distance within the distribution observed in nocodazole-treated cells, including 3 cells with inter-KT distances below 0.80 μm (Fig. 3f). Based on these subpopulations of BAL27862-treated cells we conclude that the SAC can be satisfied despite reduced MT occupancy and complete absence of inter-KT stretching.

Previous studies have shown that the SAC can act like a rheostat, in which mild KT–MT attachment defects, such as a single unattached KT, will not permanently block anaphase onset, but rather lead to a transient delay[46,47]. Since BAL27862 treatment led to a small delay in anaphase onset, it raised the possibility that partial MT occupancy and low inter-KT distances might lead to a "weak" checkpoint response that would marginally delay anaphase onset. To test this, we used hTert-RPE1 cells expressing endogenously tagged Venus-Mad2, a marker which reflects the output of the SAC by being recruited to unattached KTs[46]. We stained these cells with SiR-Hoechst, a live cell dye for DNA[48], and monitored Venus-Mad2 levels, as cells progressed through mitosis (Fig. 4a). In both DMSO- and BAL27862-treated cells, unattached KTs recruited the same levels of Venus-Mad2 (Fig. 4a, b and b). Venus-Mad2 also fully disappeared from aligned KTs in both conditions, even though a majority of KTs in BAL27862-treated cells have a reduced MT occupancy and low inter-KT distances (Figs. 2 and 4a). Finally, we observed in both conditions the same 5 min delay between the final disappearance of Mad2 and anaphase onset ($\pm 1.29$ min s.d for DMSO and $\pm 1.17$ min s.d. for BAL27862; $N = 4$; Fig. 4c). These data confirmed that BAL27862 does not impair SAC signalling and does not affect the activity of the anaphase-promoting complex (APC/C) once the checkpoint is satisfied. Furthermore, they indicate that reduced MT occupancy and low inter-KT distances do not lead to a

**Fig. 2** MT occupancy controls inter-KT stretching. **a** Schematic of human KTs showing relative positions of GFP-CENPA (green) and Ndc80 (9G3, red). **b** Immunofluorescence images of hTert-RPE1 GFP-CENPA cells (green) treated with DMSO or 15 nM BAL27862 stained for Ndc80 (red). Insets illustrate sub-pixel localization of GFP (green crosses) and Ndc80 (red crosses). Scale bars = 2 μm for merged images, 500 nm for insets. **c** Schematic of inter- and intra-KT distances and swivel measurements. **d** Quantification of inflation-corrected 3D intra-KT distances for cells treated with DMSO or 15 nM BAL27862. $N = 4$, $n = 2251$–2507 KTs, $p = 0.013$ in two-tailed $z$-test. Values given are means ± s.e.m. **e** Swivel ranges for KTs in cells treated with DMSO (black), or 15 nM BAL27862 (red). Values given are s.d., $N = 4$; $n = 1952$–2263 sister-KT pairs. **f** Autocorrelation curves of sister-KT pairs of cells treated with DMSO (black), 12 nM BAL27862 (red), or 1 μg ml$^{-1}$ nocodazole (blue); yellow is 95% CI for BAL27862-treated cells; $n = 770$-880 sister-KT pairs; $N = 2$-4. **g** Distribution of inter-KT distances (CENPA-CENPA) in cells treated with DMSO (black), 12 nM BAL27862 (red) or 1 μg ml$^{-1}$ nocodazole (blue), $N = 2$-4; $n = 770$–880 sister-KT pairs; values are means and s.d.; all conditions are different from one another ($p < 10^{-4}$, two-tailed Mann–Whitney $U$-test). **h** Resting inter-KT distances measured in cells co-treated with 1 μg ml$^{-1}$ nocodazole and either DMSO (black, $n = 289$) or 12 nM BAL27862 (red, $n = 310$; values are means and s.d.). **i** Sister-KT velocities of cells treated with DMSO, 12 nM of BAL27862, or 1 μg ml$^{-1}$ nocodazole. $N = 2$-4; $n = 549$–579 sister-KT pairs; values are means and s.e.m., $p = 0.007$ DMSO vs. BAL27862, $p = 0.002$ BAL27862 vs. nocodazole in two-tailed ANOVA test; ** represents $p < 0.01$. **j** Distribution of inter-KT distances in the subpopulation of BAL27862-treated cells with average inter-KT distance <0.8 μm (red; $n = $ 182 sister-KTs; $N = 2$-4; values are means and s.d.). **k** Autocorrelation curves and 95% CI (yellow) of sister-KT pairs shown in **j**

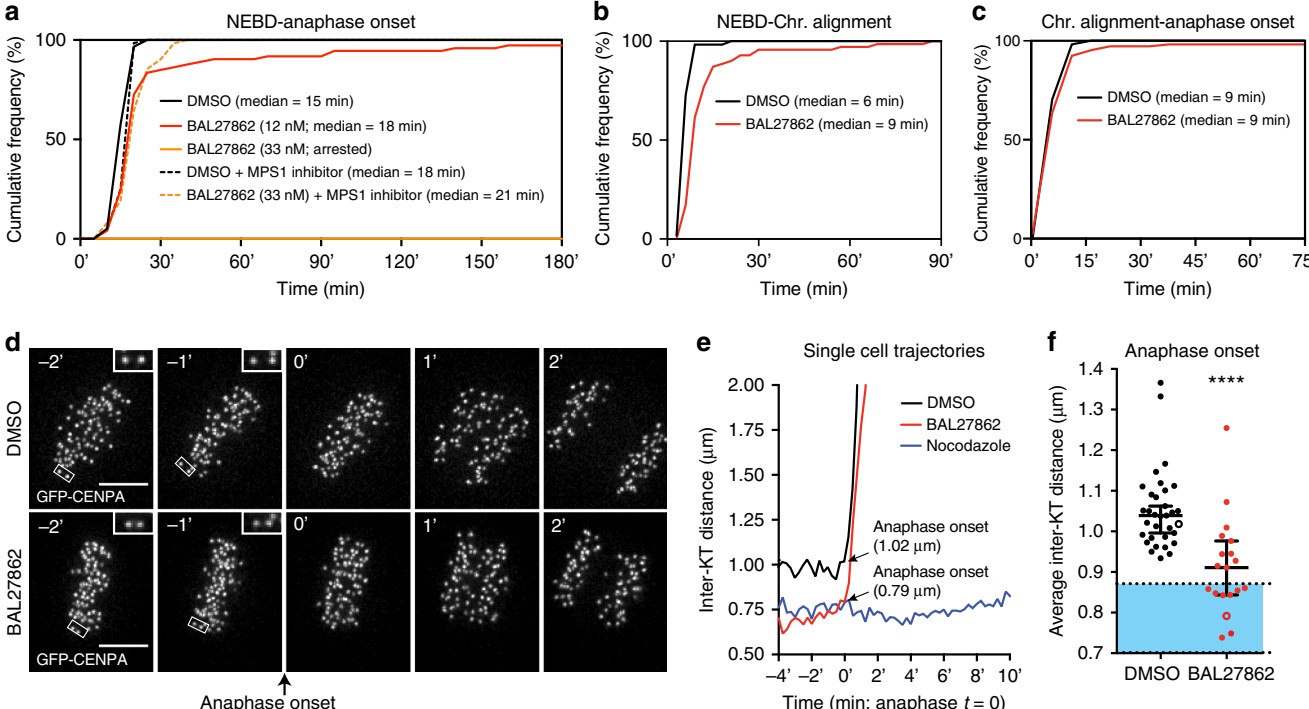

**Fig. 3** Reduced MT occupancy does not prevent the mitotic progression. **a** Cumulative frequency graph of anaphase onset time in hTert-RPE1 H2B-mCherry/EB3-eGFP cells treated with DMSO (black), 12 nM (red), 33 nM BAL27862 (orange), DMSO+1 µM reversine (dashed black) or 33 nM BAL27862 +1 µM reversine (dashed orange). $T = 0$ was set at nuclear envelope breakdown. $N = 3$, $n = 57$–74 cells for experiments without reversine. $N = 2$; $n = 62$–64 for reversine experiments. For the reversine experiments chromosome decondensation defined mitotic exit, since anaphase often failed. **b**, **c** Cumulative frequency graph of chromosome alignment time (**b**), and the time between chromosome alignment and anaphase onset (**c**) in hTert-RPE1 H2B-mCherry/EB3-eGFP cells treated with DMSO (black) or 12 nM BAL27862 (red). $T = 0$ was set either at nuclear envelope breakdown (**b**) or once all chromosomes were aligned (**c**). $N = 3$, $n = 57$–74 cells. **d** Time lapse images of hTert-RPE1 GFP-CENPA cells before and after anaphase onset ($T = 0$). Scale bars = 5 µm. **e** Exemplary curves of average inter-KT distances at anaphase onset in hTert-RPE1 GFP-CENPA cells treated with DMSO (black), 12 nM BAL27862 (red) or 1 µg ml$^{-1}$ nocodazole (negative control; blue). Time is measured in minutes using anaphase onset as $T = 0$ for DMSO or BAL27862-treated cells. **f** Median inter-KT distances at anaphase onset measured in single hTert-RPE1 GFP-CENPA cells treated with DMSO (black dots, $n = 33$ cells) or 12 nM BAL27862 (red dots, $n = 19$ cells; $N = 4$; error bars represent 95% CI, $p < 10^{-4}$ two-tailed Mann–Whitney test); the blue area represents the range comprising 80% of all inter-KT distances in nocodazole-treated cells. The empty dots indicate the cells shown in **e**. **** represents $p < 0.0001$

"weak" SAC response. Rather, we suspect that the delay in anaphase onset seen in BAL27862-treated cells might be caused by an inefficient initial capture of KTs, as reflected by the 3 min delay in chromosome alignment (Fig. 3b).

The Mps1 kinase, the most upstream SAC component, competes with MTs to bind the Ndc80 complex, the main MT-binding site on KTs[15,16]. This raised the question as to how partial MT occupancy can satisfy the SAC. We reasoned that either KTs possess individual MT-binding units that still load subcritical levels of Mps1 in BAL27862-treated cells, or that Mps1 binding to Ndc80 at individual KTs is controlled in a cooperative manner. To differentiate between the two possibilities, DMSO-, BAL27862-, or nocodazole-treated cells were stained with Mps1 antibodies and their levels at KTs quantified by indirect immunofluorescence (all cells were also treated with MG132 to ensure bi-orientation in DMSO- and BAL27862-treated cells; Fig. 4d). Mps1 levels at KTs were high after nocodazole treatment, but undetectable in both DMSO- and BAL27862-treated cells (Fig. 4e). We conclude that partial MT occupancy is sufficient to strip Mps1 away from Ndc80 complexes at KTs, arguing in favour of a cooperative binding model. One attractive possibility for such a cooperative mechanism is the Ndc80 "lawn" model[40], which proposes that individual Ndc80 complexes within a KT form a "lawn", in which fewer MTs bind a higher number of Ndc80 molecules.

**Low KT stretching barely activates Aurora-B at KTs.** Sister-KTs with low inter-KT distances, such as syntelically attached sister-KTs, are thought to undergo error correction, mediated by Aurora-B phosphorylation of KT components, such as the Mis12 and Ndc80 complexes, in a distance-dependent manner[26]. Even though RPE1 cells treated with low doses of BAL27862 have low inter-KT distances, there was no indication of an Aurora-B-driven detachment of KT–MTs, as cells maintained robust bi-oriented attachments and rapidly entered anaphase (Figs. 2 and 3). This prompted us to quantify Aurora-B activity at KTs in BAL27862-treated hTert-RPE1 GFP-CENPA cells, using two phospho-antibodies raised against key Aurora-B substrates as read-outs: pSer100 of Dsn1 (Mis12 complex[27]) and pSer44 of Ndc80 (Ndc80 complex[28]). Consistent with previous studies, Aurora-B activity was high on sister-KTs in cells treated with the Eg5-inhibitor monastrol (Fig. 5a and Supplementary Fig. 1a), as this leads to monopolar spindles that favour the formation of syntelic sister-KTs[49]. In contrast, lower activity was measured in untreated cells with bipolar spindles (Fig. 5a, and Supplementary Fig. 1a). In both cases the kinase activity was abolished following treatment with the Aurora-B inhibitor ZM1 (Fig. 5a and Supplementary Fig. 1a)[50].

BAL27862 did not weaken Aurora-B activity as monastrol-treated hTert-RPE1 GFP-CENPA cells treated with or without low doses of BAL27862 had equally high levels of phospho-Dsn1

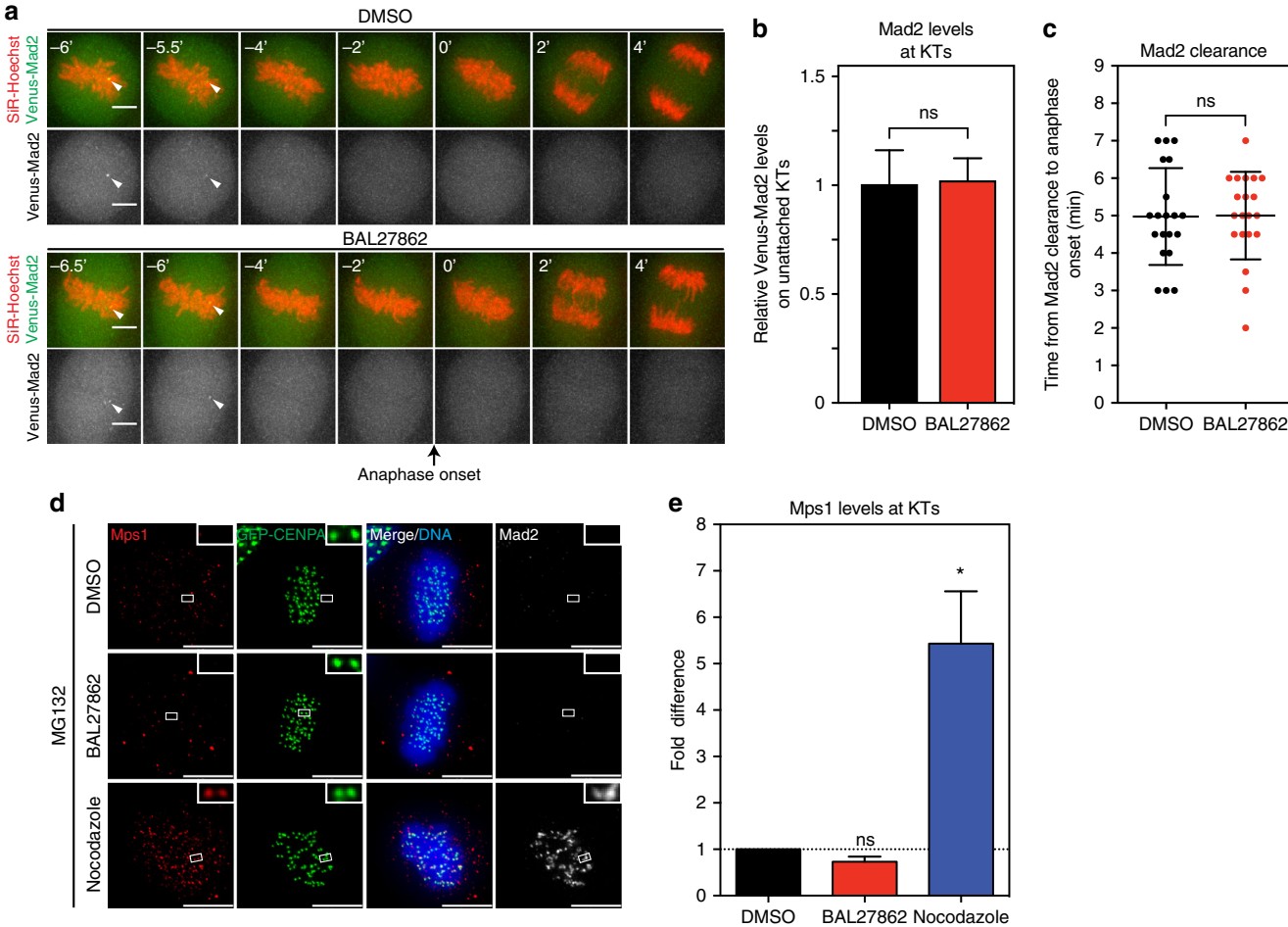

**Fig. 4** Cells satisfy the SAC despite low MT occupancy. **a** Time lapse images of hTert-RPE1 Venus-Mad2 cells around anaphase onset (black arrow) treated with DMSO or 15 nM BAL27862. Chromosomes were visualized with SiR-Hoechst (red). White arrowheads indicate the Venus-Mad2 signal on the last unattached KT. **b** Quantification of Venus-Mad2 levels on unattached KTs in hTert-RPE1 Venus-Mad2 cells. N = 3, n = 51–58 unattached KTs; error bars indicate s.e.m. **c** Quantification of the time between the disappearance of the last Mad2 signal and anaphase onset, each dot represents one cell, N = 3-4, n = 20 cells, error bars represent s.d. of means, p = 0.95, two-tailed t-test, ns, not significant. **d** Immunofluorescence images of hTert-RPE1 GFP-CENPA cells (green) treated with DMSO or 12 nM BAL27862, blocked in metaphase with 10 µM MG132, and stained with Mps1 (red) and Mad2 (white) antibodies; insets show exemplary sister-KT pairs. **e** Quantification of Mps1/GFP-CENPA ratio after background subtraction in images as shown in **d**, error bars indicate s.e.m., p = 0.012 in two-tailed ANOVA test, *significantly different in multiple comparison p = 0.038, N = 3, n = 320–700 KTs, ns, not significant

(Fig. 5b). In contrast, in metaphase cells that entered mitosis in the presence of BAL27862 (4 h treatment), both phospho-Dsn1 and phospho-Ndc80 levels were low, despite the fact that these sister-pairs never experienced high inter-KT distances (Fig. 5c and Supplementary Fig. 1b). Nevertheless, BAL27862 led to a uniform but mild (20–30%) increase in both phospho-Dsn1 and phospho-Ndc80 levels when compared to DMSO-treated metaphase cells (Fig. 5c and Supplementary Fig. 1b). We conclude that Aurora-B responds to low inter-KT distances in a manner that is insufficient to detach KTs.

**Complete MT occupancy favours segregation of merotelics.** The absence of Aurora-B-dependent detachments of sister-KTs with low inter-KT distances raised the question as to what extent Aurora-B-dependent error correction still functions in BAL27862-treated cells. To quantify error correction, we treated cells with monastrol to generate monopolar spindles that accumulate syntelic KT–MT attachments, before washing out monastrol[49]. Time lapse imaging allowed quantification of the time required for cells to correct syntelic attachments, align their

chromosomes and progress to anaphase after monastrol release, a read-out of the Aurora-B-dependent error correction efficiency[49] (Fig. 6a). We observed no difference in chromosome alignment and anaphase timing upon monastrol release (Fig. 6b, c), indicating that BAL27862 does not impair the ability of Aurora-B to correct syntelic attachments. Moreover, it demonstrated that low inter-KT distances and reduced MT occupancy do not change the kinetics of SAC satisfaction, in contrast to what has been proposed[51] and it confirmed our hypothesis that the mild anaphase delay seen in BAL27862-treated cells (Fig. 3a, b) is due to the inefficient initial capture of KTs.

The formation of a transient monopolar spindle not only induces formation of syntelic attachments, but also leads to lagging chromosomes. About 1–2% of these lagging chromosomes are due non-disjoined chromosomes[35], but the rest is due to merotelic attachments, in which one KT is bound by MTs from both spindle poles[52]. Since sister-KTs with a merotelic attachment are still stretched apart, their correction is thought to be independent of inter-KT stretching[13]. Nevertheless, BAL27862 treatment induced a modest but highly reproducible increase in the percentage of cells with lagging chromosomes

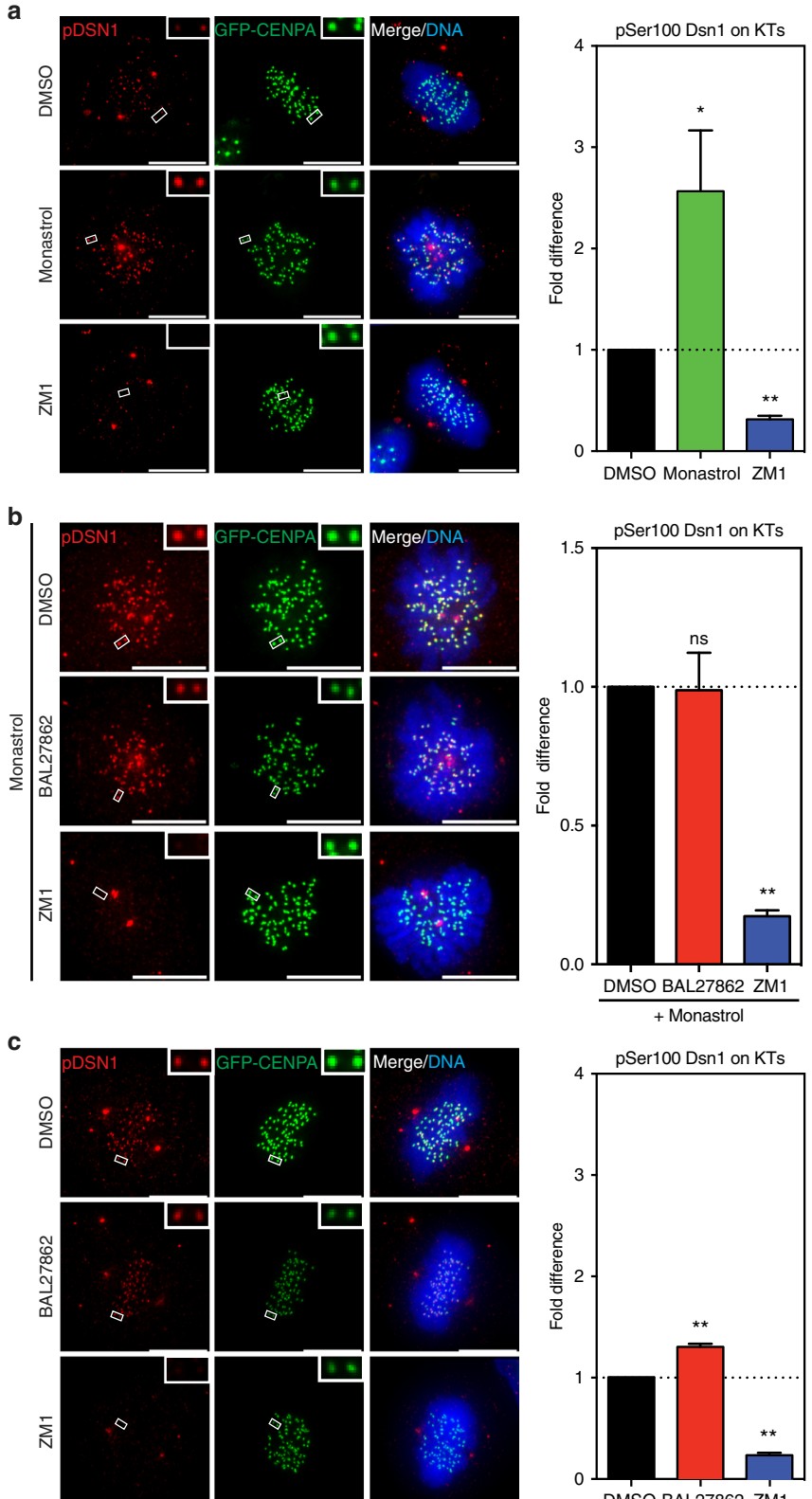

(29 ± 3.1% vs. 20% ± 2.7% in DMSO-treated cells; error bars indicate s.e.m; note that the rare cases of non-disjoined chromosomes, in which two lagging chromosome masses were visible, were excluded from this analysis; $N = 9$; $p = 0.0135$ in paired $t$-test; Fig. 6d). Immunofluorescence of fixed cells 60 min after a monastrol release confirmed the frequent presence of lagging chromosomes with merotelic KT–MT

attachments (Fig. 6e). Overall, this suggested that reduced MT occupancy lowers the efficiency of merotelic correction.

**Complete KT–MT occupancy is needed for robust anaphase A.** Stabilization of KT–MTs is thought to be the major source of merotelic attachments due to impaired error correction[13].

Nevertheless, when we tested in hTert-RPE1 GFP-CENPA cells the effects of low doses of BAL27862 on KT-MTs using a cold stable assay, we rather observed a mild decrease in MT stability (Fig. 7a, b). This excluded a priori the possibility that the increased incidence in lagging chromosomes is caused by MT stabilization.

An alternative hypothesis is based on the fact that merotelic KTs are bound to more MTs from the "correct" half-spindle than

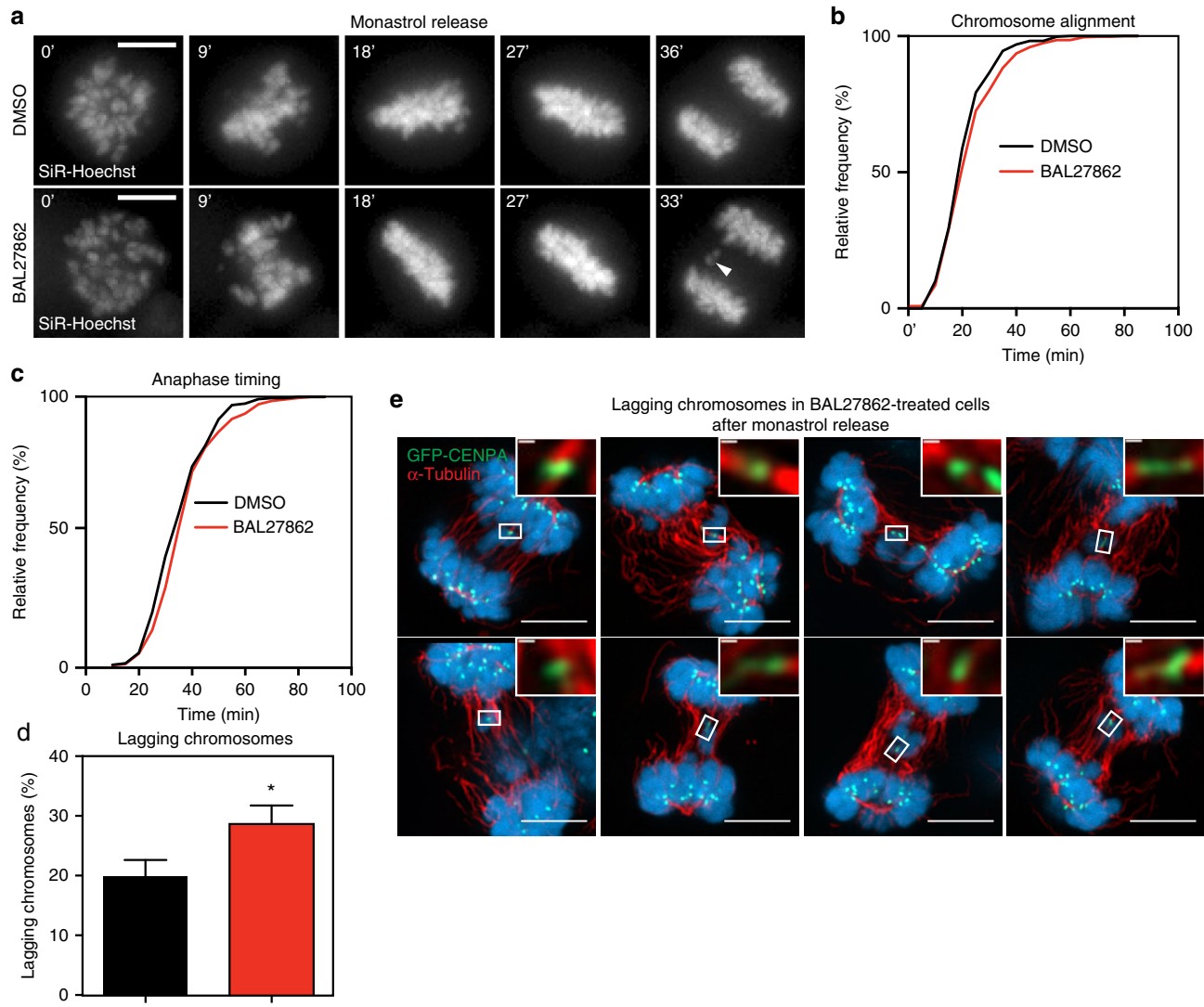

**Fig. 6** Complete KT–MT occupancy prevents lagging chromosomes. **a** Time lapse images of hTert-RPE1 GFP-CENPA cells released from a monastrol-treatment ($T = 0$ min). SiR-Hoechst was used to visualize chromosomes; a lagging chromosome is marked with white arrowhead. Scale bar = 5 µm. **b**, **c** Cumulative frequency graph of chromosome alignment (**b**) and anaphase time (**c**) in hTert-RPE1 GFP-CENPA cells treated with DMSO (black) or 12 nM BAL27862 (red) in a monastrol-release assay. $T = 0$ at monastrol release. $N = 9$, $n = 530$–549 cells. **d** Percentage of cells treated with DMSO or 12 nM BAL27862 with lagging chromosomes in anaphase after a monastrol release, * DMSO vs BAL27862 $p = 0.0135$ in ratio paired two-tailed $t$-test, $N = 9$, $n = 459$–469 cells, error bars show s.e.m. **e** Immunofluorescence images of 10 hTert-RPE1 GFP-CENPA cells from a single monastrol release experiment. Cells were treated in parallel with DMSO or 12 nM BAL27862 and 100 µM monastrol (4 h), released from monastrol for 60 min, fixed and stained with α-tubulin antibodies (red). Scale bars = 5 µm. Inserts show single, merotelically attached KTs. Scale bars = 250 nm

**Fig. 5** Low inter-KT stretching barely activates Aurora-B at KTs. **a** Left panel shows immunofluorescence images of hTert-RPE1 GFP-CENPA cells (green) treated with DMSO, 100 µM monastrol, or 2 µM Aurora-B inhibitor ZM1, and stained with DAPI (blue) and pSer100-Dsn1 antibodies (red); insets show exemplary sister-KT pairs; right panel shows quantification of normalized pSer100-Dsn1/GFP-CENPA ratio after background subtraction. *DMSO vs. monastrol $p = 0.0265$, **DMSO vs. ZM1 $p = 0.0023$ in ratio paired two-tailed t-test, error bars represent s.e.m., $N = 4$, $n = 340$–500 KTs. **b** Left panel shows immunostaining of pSer100-Dsn1 in hTert-RPE1 GFP-CENPA cells co-treated with 100 µM monastrol, and DMSO, 12 nM BAL2782 or 2 µM ZM1; insets show exemplary sister-KT pairs; right panel shows quantification of normalized pSer100-Dsn1/GFP-CENPA ratio after background subtraction. DMSO vs. BAL27862 $p = 0.8442$, **DMSO vs. ZM1 $p = 0.0046$, in ratio paired two-tailed $t$-test, error bars represent s.e.m., $N = 3$, $n = 600$ KTs. **c** Left panel shows immunostaining of pSer100-Dsn1 (red) in metaphase hTert-RPE1 GFP-CENPA cells treated with DMSO, 12 nM BAL27862, or 2 µM ZM1; insets show exemplary sister-KT pairs; right panel shows quantification of normalized pSer100-Dsn1/GFP-CENPA ratio after background subtraction, **DMSO vs. BAL27862 $p = 0.0072$, **DMSO vs. ZM1 $p = 0.0048$ in paired two-tailed $t$-test, error bars represent s.e.m., $N = 3$, $n = 300$–600 KTs

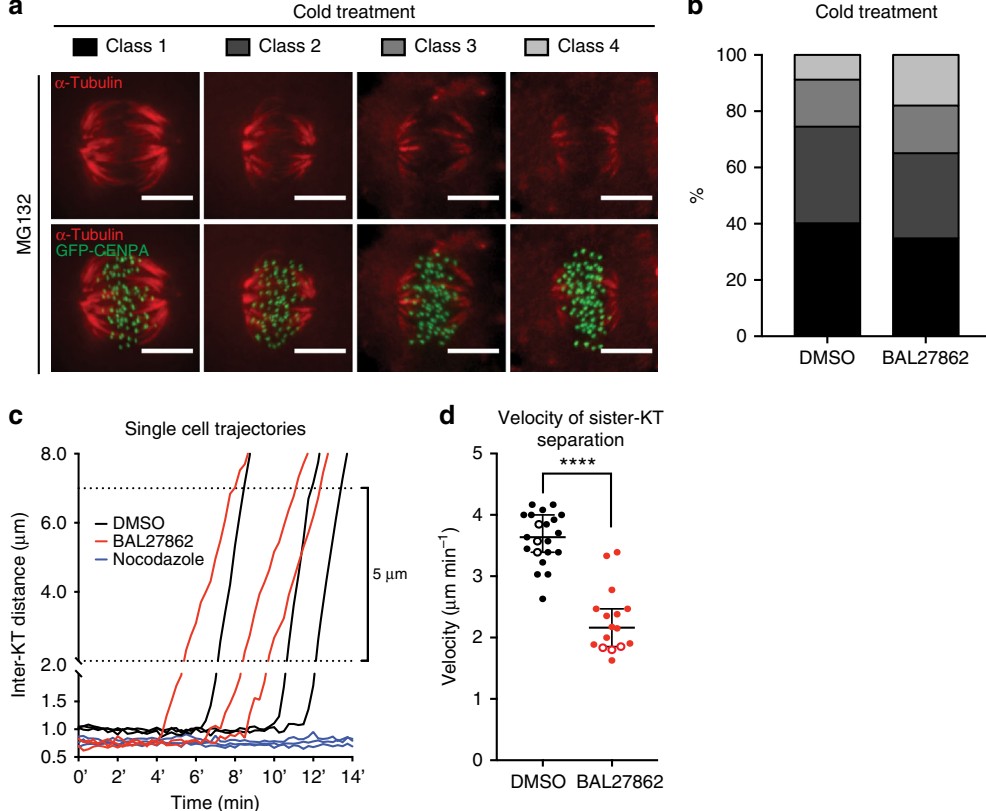

**Fig. 7** Complete KT–MT occupancy ensures robust anaphase forces. **a**, **b** Cold stable assay in hTert-RPE1 GFP-CENPA cells treated with DMSO or 12 nM BAL27862 and stained for α-tubulin. **b** Quantification of the cold stable assay. $N = 3$, $n = 89$–$102$ cells. **c** Exemplary curves of average sister-KT distances in anaphase in hTert-RPE1 GFP-CENPA cells treated with DMSO (black), 12 nM BAL27862 (red) or 1 µg ml$^{-1}$ nocodazole (negative control; blue). $T = 0$ indicates the start of the movies. **d** Dot plot of anaphase speeds in hTert-RPE1 GFP-CENPA cells treated with DMSO or 12 nM BAL27862; empty dots are cells represented in **c**; $N = 3$, $n = 16$–$21$, error bars show 95% CI of medians, **** represents $p < 0.0001$ in two-tailed $t$-test

the "incorrect" one, resulting in a force differential[53,54]. This force differential can correct merotelic laggards in anaphase, by segregating merotelically attached KTs towards the correct pole[53,54]. Since the number of KT–MTs on the "incorrect" side of a sister-KT can vary, a high occupancy might ensure that there is always a surplus of MTs on the "correct" side, to favour segregation towards the correct pole. Conversely, low MT occupancy might increase the probability of a force balance, and thus prevent the anaphase-driven error correction. To test this hypothesis we inferred the force acting on KTs in early anaphase by quantifying the speed at which sister-KTs segregate from one another using hTert-RPE1 GFP-CENPA cells (Fig. 7c):[55] sister-KT separation was slowed down by 41% or 1.5 µm min$^{-1}$ in BAL27862-treated cells when compared to DMSO-treated cells (median $3.63 \pm 0.38$ µm min$^{-1}$ 95% CI for DMSO and $2.16 \pm 0.55$ µm min$^{-1}$ for BAL27862; $N = 3$; Fig. 7c, d). We conclude that BAL27862 reduces anaphase velocity.

Anaphase sister-KT separation is not only driven by KT–MT generated forces (anaphase A), but also by spindle elongation (anaphase B), which is driven by pulling forces on MTs and MT sliding at inter-polar MTs[56,57]. Immunofluorescence microscopy revealed that a low nanomolar BAL27862 treatment led to smaller asters at the poles (Supplementary Fig. 2a) and a 24.5% reduction in PRC1 intensity, a marker for inter-polar MTs[58] (±3.3% based on 95% CI; $N = 3$; Fig. 8a, b). Nevertheless, when we performed 4D KT and spindle pole-tracking in hTert-RPE1 Centrin1-GFP/GFP-CENPA cells we found no change in spindle elongation rates between DMSO and BAL27862-treated cells, even though average inter-KT distances were low (Fig. 8c, d; Supplementary

Fig. 2b and c). We conclude that the reduction in anaphase forces is only due to changes in anaphase A.

Finally, anaphase A itself is driven by both plus-end and minus-end MT depolymerization. Minus-end depolymerization at spindle poles gives rise to poleward MT flux, which has been shown to contribute to up to 20% of anaphase speed[59]. We therefore performed a photoactivation assay using hTert-RPE1 PA-GFP-α-tubulin cells to measure poleward MT flux in BAL27862-treated cells. We found that flux rates decreased by 0.4 µm min$^{-1}$ ($0.69 \pm 0.11$ µm min$^{-1}$ s.e.m; $N = 3$) when compared to DMSO ($1.09 \pm 0.11$ µm min$^{-1}$ s.e.m.; $N = 3$; Fig. 8e and f). Since the reduction in anaphase A speed is much larger (1.5 µm min$^{-1}$; Fig. 7d), we conclude that the reduction in flux rate only mildly contributes to reduced anaphase A forces. Overall, we conclude that a reduction in MT occupancy diminishes the forces acting on KTs in early anaphase, and they provide support for our model that a reduction in MT occupancy diminishes the probability of a robust force differential at merotelically attached KTs (Fig. 9a,b).

## Discussion

Our data indicate that the segregation of merotelically attached kinetochores is favoured by high microtubule occupancy, which ensures a strong anaphase A force. The usage of nanomolar concentrations of BAL27862 furthermore gave us key new insights into how forces affect chromosome movements and the KT structure itself. Finally, we demonstrate that high MT occupancy and the resulting inter-KT stretching is neither required for

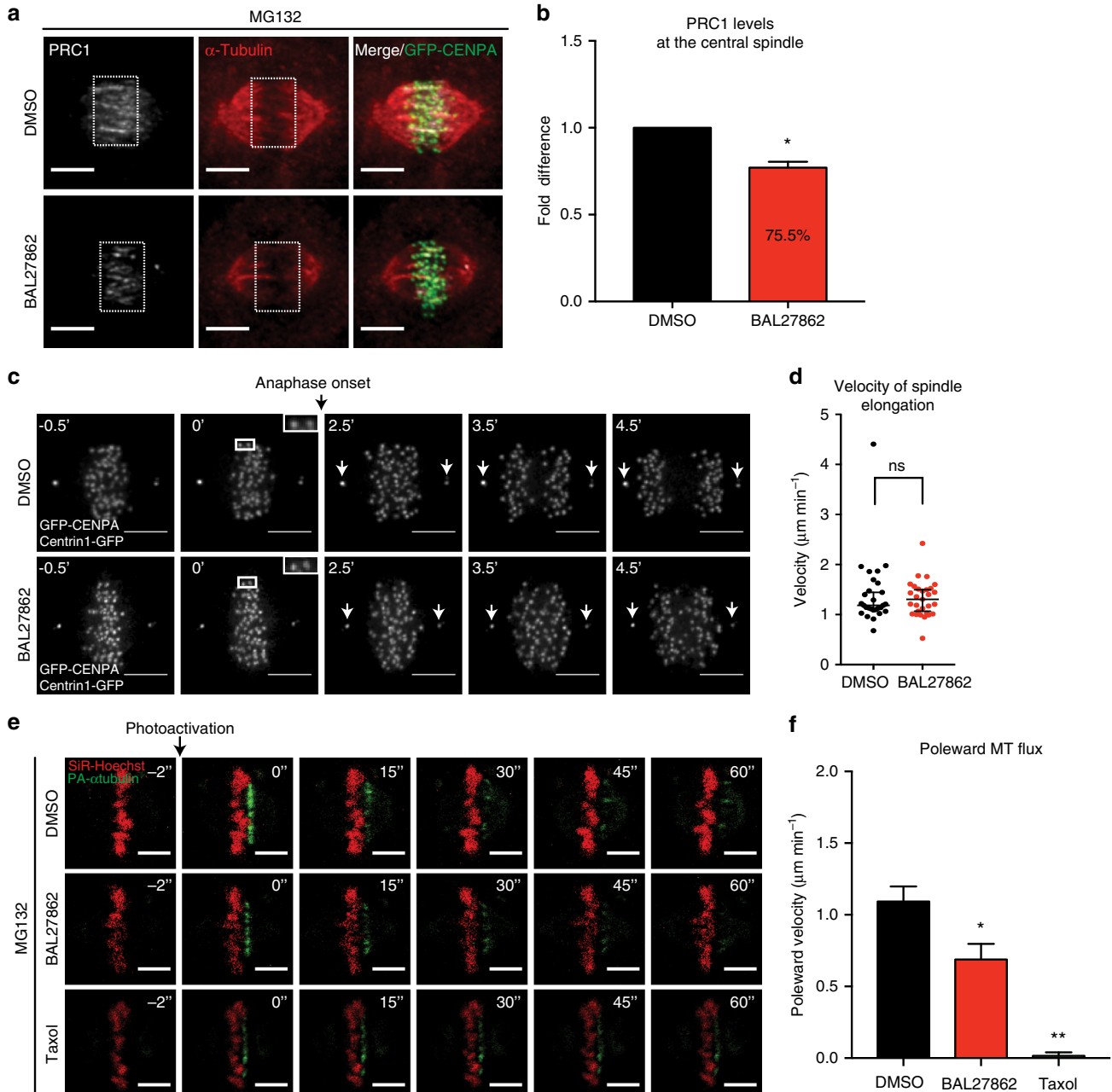

**Fig. 8** Low doses of BAL27862 specifically affect anaphase A. **a** Immunofluorescence images of hTert-RPE1 GFP-CENPA cells treated in parallel with DMSO or 12 nM BAL27862, and 10 μM MG132 after staining for α-tubulin (red) and PRC1 (grey). Scale bars = 5 μm. **b** Quantification of PRC1 intensities in hTert-RPE1 GFP-CENPA cells normalized to DMSO; $N = 3$, $n = 104–105$ cells, error bars are s.e.m, * represents $p = 0.026$, ratio paired two-tailed $t$-test. **c** Time lapse images of hTert-RPE1 Centrin1-GFP/GFP-CENPA cells treated with DMSO or 12 nM of BAL27862 (4 h). Scale bars = 5 μm. White arrows at spindle poles indicate position of centrosomes (Centrin1-GFP). **d** Quantification of spindle elongation velocities of hTert-RPE1 Centrin1-GFP/GFP-CENPA cells treated with DMSO or 12 nM BAL27862; $N = 4$, $n = 28–29$ cells, error bars are 95% CI of medians, $p = 0.968$, two-tailed Mann–Whitney test. **e** Time lapse images of hTert-RPE1 PA-GFP-α-tubulin cells treated in parallel with 25 nM SiR-Hoechst (red); DMSO, 12 nM BAL27862, or 10 μM taxol; and 10 μM MG132. The green signal indicates the position of the PA-GFP-α-tubulin over time. Scale bars = 5 μm. **f** Quantification of poleward MT flux rates of hTert-RPE1 PA-GFP-α-tubulin cells; $N = 3$, $n = 30–49$ k-fibres from 17 to 30 cells, error bars are s.e.m., * represents $p = 0.041$ DMSO vs. BAL27862; ** represent $p = 0.004$ for BAL27862 vs. taxol, in two-tailed ANOVA test

SAC silencing nor is it involved in the detachment of syntelic MT attachments by Aurora-B (Fig. 9a).

Our data show that BAL27862 treatment cells slow down anaphase A movements by 1.5 μm min⁻¹, without affecting spindle elongation (anaphase B). Assuming that at the sub-cellular levels viscous forces dominate, and that the velocity of chromosome movements are directly proportional to the forces acting on them[55], we infer a 40% reduction in the forces

pulling on chromosomes. Our experiments point to two causes for this force reduction: a minor contribution of poleward MT flux, which is reduced by 0.4 μm min⁻¹, and a major contribution from the reduction in KT–MT occupancy. The minor contribution of poleward MT flux is consistent with experiments showing that abrogation of poleward MT flux in human cells only leads to a mild reduction in anaphase velocity[59].

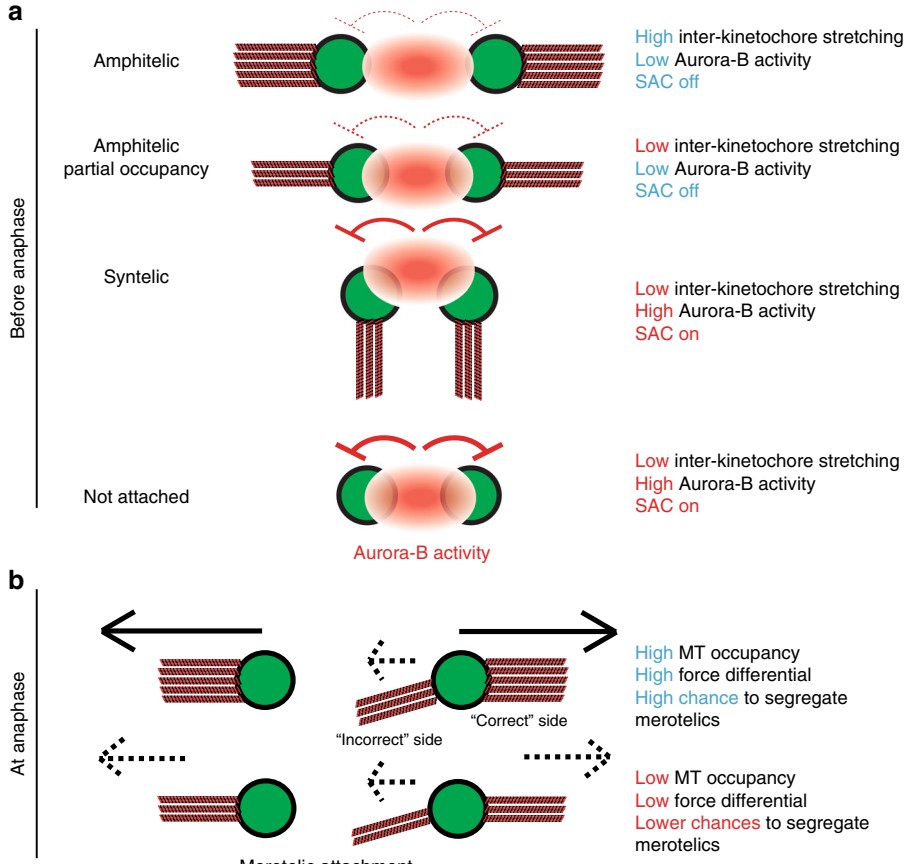

**Fig. 9** Speculative model for the role of complete KT–MT occupancy. **a**, **b** Schematic model of the impact of reduced KT–MT occupancy on sister-KTs before (**a**) and after (**b**) anaphase onset. KTs are in green, Aurora-B activity gradient in red, and MTs in red-black. Full and dashed "inhibition" signs in red represent the respective strength of Aurora-B activity on KTs. Full and dashed black arrows represent the respective strength of the force applied by MTs on sister-KTs at anaphase. We postulate that before anaphase (**a**) complete MT occupancy and high inter-KT stretching is not required for SAC satisfaction; moreover, low inter-KT distances are not sufficient to induce Aurora-B-dependent MT detachment. This indicates that Aurora-B must recognize additional features in syntelic KT pairs. After anaphase onset (**b**), complete MT occupancy favours the segregation of merotelic chromosomes by increasing the likelihood of a high force differential between the "correct" and "incorrect side of the lagging chromosome

Based on our anaphase velocity measurements, we postulate that the reduction in anaphase A forces decreases the likelihood of a high force differential between the correct and incorrect side of a merotelically attached KT. This increases the chances that a merotelic chromosome remains in the middle of the spindle for a long time, being caught in a tug of war of equal pulling forces (Fig. 9b). Importantly, even if such a chromosome is ultimately segregated, it will form genetically unstable micronuclei[60]. We postulate that the high number of KT–MTs ensures a robust anaphase A force to promote efficient chromosome segregation.

The treatment of RPE1 cells with a low nanomolar concentration of BAL27862 also provides insights into the mechanics of sister-KT movements. Under these conditions the sister-KTs move more slowly and do not stretch their centromeres (low inter-KT distances); nonetheless, sister-KTs were still able to undergo oscillatory motion along the spindle axis, albeit with a shorter period (Fig. 2). This suggests that high inter-KT stretching of the centromere is not essential to drive KT directional switching and points to small changes in the spring force being sufficient to regulate switching events[61]. A second striking finding is that a reduction in the force acting on sister-KTs led to no change or even to an increase in intra-KT distances (it is difficult to distinguish between the two, as the increase we saw is close to the known methodological uncertainty). This further supports the model that KTs are non-compliant structures[42,62], and that

changes in intra-KT distances most likely reflect conformational changes in response to attachment status, rather than a force-dependent stretching of the KT itself.

Our data further directly show that inter-KT stretching is not required for rapid SAC satisfaction of wild-type KTs, corroborating previous studies that used KTs with non-phosphorylatable Ndc80 mutants[21,22]. Moreover, we demonstrate that KTs that have ~2/3rd of MT-binding sites occupied satisfy the SAC and do not load Mps1. These data are in agreement with the Ndc80 complexes "lawn" model, in which MTs bind to different numbers of Ndc80 complexes depending on MT occupancy[40]. Furthermore, it would indicate that cells do not "check" if KTs reach full MT occupancy, but rather rely on the time window between checkpoint satisfaction and anaphase onset to fully mature their KT fibres.

The final, major implication of our data is that low inter-KT distances do not induce Aurora-B-dependent error correction. Our data are consistent with the presence of an Aurora-B activity gradient, as we find that low inter-KT distances enhance phosphorylation of the Mis12 and Ndc80 complexes, but the increase in phosphorylation is only modest, and not sufficient for MT detachment. This indicates that in chromosomes with syntelic attachments, the low inter-KT distances and the Aurora-B activity gradient do not suffice to explain error correction, and that alternative or additional mechanisms must exist (Fig. 9a).

One possibility is the inherent geometric deformation of the centromeric DNA in syntelic KT–MT attachments, which could enhance Aurora-B activity. A second possibility is that Aurora-B synergistically cooperates with centrosomal Aurora-A, which destabilizes KT–MTs in the vicinity of spindle poles[63–65]. Such a cooperative mechanism might be necessary to discriminate between syntelically attached chromosomes, which are located close to spindle poles, and the bi-oriented sister-KTs on the metaphase plate. Indeed, the inter-KT distances in bi-oriented KT pairs are not static but "breathing", with a period of 45–60 s[45], meaning that even bi-oriented sister-KT pairs remain close for extended periods of time. Therefore, relying purely on inter-KT distances to identify erroneous KT–MT attachments would be counter-productive, as it would lead to a high rate of false positives.

## Methods

**Cell culture and drug treatments**. hTert-RPE1 GFP-CENPA[45], hTert-RPE1 Centrin1-GFP/GFP-CENPA (kind gift of A. Khodjakov), and hTert-RPE1 H2B-mCherry/EB3-eGFP (kind gift of W. Krek) cell lines were cultured in Dulbecco's modified Eagle's medium (DMEM; Thermofisher, Switzerland) supplemented with 10 fetal calf serum (FCS), 100 U ml$^{-1}$ penicillin, and 100 mg ml$^{-1}$ streptomycin. hTert-RPE1 eGFP-α-tubulin (kind gift of D. Gerlich) and hTert-RPE1 PA-GFP-α-tubulin cells[66] were cultured in DMEM (Thermofisher, Switzerland) medium supplemented with 10% FCS, 100 U ml$^{-1}$ penicillin, and 100 mg ml$^{-1}$ streptomycin and either with 0.5 µg ml$^{-1}$ puromycin (Life Technologies; hTert-RPE1 GFP-α-tubulin) or 500 µg ml$^{-1}$ G418 (Life Technologies; hTert-RPE1 PA-GFP-α-tubulin). hTert-RPE1 Venus-Mad2 cells (kind gift of J. Pines) were cultured in DMEM/F12 Ham with 3151 mg l$^{-1}$ dextrose, 2.5 mM L-glutamine, 15 mM HEPES, 55 mg l$^{-1}$ sodium pyruvate (Sigma Aldrich) containing 10% FCS, 100 U ml$^{-1}$ penicillin, 100 mg ml$^{-1}$ streptomycin, and 0.25 µg fungizone (Bioconcept, Switzerland). HeLa HaloTag-CENPA cells[42] were cultured in DMEM supplemented with 10% FCS, 100 U ml$^{-1}$ penicillin, 100 mg ml$^{-1}$ streptomycin, and 300 µg ml$^{-1}$ G418. All cells were cultured in 37 °C with 5% CO$_2$ in humidified incubator. Live cell experiments were performed using ibidi 8-well chambers (Vitaris, Switzerland) or µ-Dish 35 mm (Vitaris) in Leibovitz L-15 (Thermofisher) medium supplemented with 10% FCS. Cells were treated with 1 µM reversine (Mps1 inhibitor; Sigma Aldrich), 100 µM monastrol (Eg5 inhibitor; Sigma Aldrich) for 4 h, 2 µM ZM1 (Aurora-B inhibitor; Enzo Life Sciences, Switzerland) for 4 h, 10 µM MG132 (proteasome inhibitor; Sigma Aldrich) for 2 h, 1 µg ml$^{-1}$ nocodazole (Sigma Aldrich) for 1–4 h, and various concentrations of BAL27862 (kindly provided by Basilea Pharmaceutica International Ltd, Switzerland) for 4 h. The concentration of BAL27862 was 12 or 15 nM, depending on the cell line. This variation was due to the fact that different RPE1 cell lines had slightly different concentrations to obtain a consistent phenotype: a reduction of at least 50% in inter-KT stretching when compared to the resting distance, no block in anaphase onset, and robust sister-KT oscillations as measured in our automated KT-tracking assay[45]. The used concentrations are indicated in the figure legends.

**Live cell imaging and KT tracking**. Cells were imaged either using a Nikon Eclipse Ti-E wide-field microscope (Nikon, Switzerland) equipped with a DAPI/eGFP/TRITC/Cy5 filter set (Chroma, USA) and a 40× NA 1.3 objective (mitotic timing) and recorded with an Orca Flash 4.0 CMOS camera (Hamamatsu, Japan) and the NIS software; an Olympus DeltaVision wide-field microscope (GE Healthcare, Switzerland) equipped with a eGFP/RFP filter set (Chroma) and with 40× NA 1.3 (monastrol release), 60× NA 1.4 (Mad2 clearance), or 100× NA 1.4 objectives (KT tracking) and recorded with a Coolsnap HQ2 CCD camera (Roper Scientific, USA) and the Softworx software (GE Healthcare). For measurements of intra-KT distances, cells were imaged using a confocal spinning-disk microscope (VOX UltraView; PerkinElmer, UK), with a 100× 1.4 NA oil objective and recorded with a Hamamatsu ORCA-R2 camera using Volocity 6.0 software (PerkinElmer). The 12 µm z-stacks were imaged with z-slices separated by 200 nm, imaged for each 488 (525) and 561 nm (615 nm) excitation (emission) wavelengths, with 50 ms exposure per z-slice. To measure mitotic timing hTert-RPE1 H2B-mCherry/EB3-eGFP were imaged every 3 min for 12 h with 2 µm z-stacks. DMSO or BAL27862 were added at the beginning of the imaging. hTert-RPE1 Venus-Mad2 cells were co-incubated for 4 h with 25 nM SiR-Hoechst (Spirochrome AG, Switzerland) and DMSO or BAL27862, and imaged every 30 s for 15 min with 0.5 µm z-stacks. Venus-Mad2 levels were calculated on unattached KT prometaphase cells after background subtraction. The decay of Venus-Mad2 levels was analysed on deconvolved images using ImageJ. For the monastrol release experiment hTert-RPE1 GFP-CENPA were imaged every 3 min for 2 h after monastrol release using 2 µm z-stacks. Lagging chromosomes in time lapse movies were scored using the IMARIS (Bitplane, Switzerland) software. All recorded images were transferred to ImageJ and mounted in Illustrator (Adobe). KT tracking of hTert-RPE1 GFP-CENPA and hTert-RPE1 Centrin1-GFP/GFP-CENPA cells was adapted based on

our previous work[67]. Briefly, to generally track KTs 3D stacks of single cells were recorded in 0.5 µm steps at a sampling rate of 15 s over a period of 15 min. To specifically measure the resting distance between sister-KTs a sampling rate of of 7.5 s for 5 min was applied. The 3D z-stacks were deconvolved using SoftWorx software and analysed in MATLAB (The MathWorks, Inc., Natick, MA, USA) with an automated KT tracking code[45,67] (the latest code is available under https://github.com/cmcb-warwick). Briefly, this gave us the frame-to-frame displacement of sister-KTs, as well as their relative position with regard to the centre of the metaphase plate. From these data, we extracted the inter-KT stretching of sister-KT pairs, and we used an autocorrelation function to reveal the oscillatory sister-KT movements along the spindle axis. KT velocities were calculated by plotting the distribution of all sister-KT displacements and calculating the standard deviation of this distribution. Anaphase velocity in single cell was extracted from the slope of the inter-KT curve between 2 and 7 µm and quantified with a modified KT-tracking software[67]. Anaphase spindle elongation velocities were calculated based on the position of the centrosomes in hTert-RPE1 Centrin1-GFP/GFP-CENPA cells during the time frame in which sister-KTs moved 7 µm apart.

**Electron microscopy**. hTert-RPE1 GFP-CENPA cells were first incubated with DMSO or BAL27862 alone for 2 h before adding 10 µM MG132 for an additional 2 h to enrich for metaphase cells. The mitotic cells were collected in 10 ml tubes using a mitotic shake-off, centrifuged at 100 × g for 5 min in a Megafuge 1.0 centrifuge (Heraeus Instrument, Switzerland) and fixed for 30 min at room temperature with culture medium supplemented with 2% of EM-grade glutaraldehyde (Sigma Aldrich). Cells were centrifuged again (100 × g for 5 min), collected into 1.5 µl Eppendorf tubes, and re-fixed without re-suspending the pellet with a 2% glutaraldehyde solution in 0.1 M sodium phosphate buffer (pH = 7.4) for additional 30 min at room temperature. Finally the pellet was centrifuged gently at 100 × g for 5 min in an Eppendorff mini-centrifuge (Eppendorff, Switzerland) and washed once with phosphate-buffered saline. Fixed cells were further treated with 2% osmium tetraoxyde in buffer and immersed in a solution of uranyl acetate 0.25% overnight[68]. The pellets were dehydrated in increasing concentrations of ethanol followed by pure propylene oxide, then embedded in Epon resin. Thin sections were stained with uranyl acetate and lead citrate and observed in a Tecnai 20 electron microscope (FEI Company, USA). MT numbers in KT fibres were calculated as reported:[38] bundles of MTs were only counted as k-fibres if they were within a 1 µm of a chromosome, if the bundle contained at least 7 MTs, and if the MTs were separated by less than 80 nm. The MT number was assessed in a blind manner.

**Antibody production**. PRC1 antibodies were raised in rabbits against the peptide CSKASKSDATSGILNSTNIQS coupled to KLH injected into rabbits (Pepceuticals). Polyclonal sera were collected according to a standard protocol.

**Immunofluorescence**. Cells were fixed with a fixation buffer (20 mM PIPES (pH = 6.8), 10 mM EGTA, 1 mM MgCl$_2$, 0.2% Triton X-100, 4% formaldehyde; Sigma Aldrich) for 7 min at room temperature. The following primary antibodies were used: mouse anti-CENPA (1:1000; Abcam ab13939), rabbit anti-MAD2 (1:1000; Bethyl A300-301A), mouse anti-α-tubulin (1:1000; Sigma Aldrich T9026), rabbit anti-α-tubulin (1:1000; Abcam ab18251), rabbit anti-PRC1 (1:500; this study), mouse anti-Ndc80$^{Hec1}$ (1:1000; Abcam 9G3), mouse anti-Mps1 (1 µg ml$^{-1}$; Abcam ab170190) and rabbit anti-pS100-Dsn1 (1:1000; kind gift of I. Cheeseman). For rabbit anti-pSer44-Ndc80 antibody (kind gift from J. DeLuca) cells were first pre-treated with the following lysis solution: PMEM buffer (60 mM PIPES; 25 mM HEPES; 10 mM EGTA; 4 mM MgSO$_4$; pH = 7.0, Sigma Aldrich) with phosphatase inhibitor Calyculin A 100 nM (Life Technologies) for 1 min in 37 °C and fixed with 4% formaldehyde in PMEM buffer for 20 min in 37 °C. For visualization of astral MTs, PRC1 measurements and cold stable assay hTert-RPE1 GFP-CENPA cells were first briefly rinsed with a cytoskeleton buffer (CB buffer; 10 mM MES, 150 mM NaCl, 5 mM MgCl$_2$, 5 mM glucose; Sigma Aldrich) and fixed with a glutaraldehyde solution in CB buffer (0.05% glutaraldehyde, 3% formaldehyde, and 0.1% Triton X; Sigma Aldrich) for 15 min at room temperature. To calculate chromatic aberration, HeLa HaloTag-CENPA cells were incubated before fixation for 15 min with growth medium containing 2.5 µM Oregon Green and 1 µM TMR (both Promega), followed by 30 min of growth medium alone. The cold stable assay was analysed in a blind manner. Cells treated with cold for 6 min were grouped in 4 distinct classes reflecting the increased loss of KT-MTs. For KT fibre fluorescence measurements of hTert-RPE1 eGFP-α-tubulin cells and for visualization of merotelic attachments in hTert-RPE1 GFP-CENPA cells, a previously described methanol fixation was used[69]. hTert-RPE1 eGFP-α-tubulin cells were pre-sorted by fluorescence-activated cell sorting to obtain a population of cells expressing similar levels of eGFP-α-tubulin protein and then treated for 4 h with 12 nM of BAL27862 and for 2 h with 10 µM of MG132 prior fixation. Cross-adsorbed fluorochrome-conjugated secondary antibodies were used according to the providers' instructions (Thermo Fisher Scientific). Three-dimensional image stacks of mitotic cells were acquired with 0.2 µm steps using a 60× NA 1.4 objective on an Olympus Delta-Vision microscope (GE Healthcare) equipped with a DAPI/FITC/TRITC/CY5 filter set (Chroma, Bellow Falls, VT) and a CoolSNAP HQ camera (Roper Scientific,

Tucson, AZ). Images were 3D-deconvolved using Softworx (GE Healthcare) and the KT protein intensities were analysed in ImageJ. For k-fibre intensities and visualization of merotelics, the three-dimensional image stacks of mitotic cells were acquired with a laser scanning LSM800 confocal microscope (Zeiss) using an Airyscan function, 63× NA 1.4 Oil DIC f/ELYRA objective with 405/488/561/640 nm lasers and Airyscan-mode optimized settings providing a final pixel size of 35 nm and 190 nm in z resolution. Images of 4 μm thickness were then 3D-deconvolved using ZEN software (Zeiss) and analysed in MATLAB (The Math-Works, Inc. Natick, MA, USA).

**Poleward MT flux measurements**. hTert-RPE1 PA-GFP-α-tubulin cells were stained with 25 nM SiR-Hoechst (Spirochrome; 4 h), DMSO or 12 nM BAL27862 (4 h) and 10 μM MG132 (2 h). As a negative control, cells were alternatively incubated with 10 μM taxol in MG132-arrested cells (30 min; Sigma Aldrich). Single focal planes were acquired using an A1r point scanning confocal microscope (Nikon), 60× 1.4 NA CFI Plan Apochromat objective. Spindles were photoactivated in the vicinity of the metaphase plate using 1 pulse of 1.9 s duration with 100% 405 nm laser power and then imaged every 5 s for 1 min. Photoactivated k-fibres were detected using an in-build "spot" function (500 μm diameter) and the centre of the mass of the metaphase plate was assigned for each frame using the "surface" function in Imaris (Bitplane) software. Spots were tracked for 60 s and the distance between each spot and the centre of the mass of the metaphase plate was computed for each frame. The flux rate was calculated as the difference between the initial position of each photoactivated spot and its final position 60 s later.

**High-resolution KT tracking**. The inner KT (eGFP-CENPA) position was detected by splitting the histogram of intensities, and localizing the spot centres using mixture model fitting (MMF) of 3D Gaussians. The outer KT (Ndc80) position was detected by finding the maximum intensity pixel within a radius of 300 nm of the inner KT position after preliminary correction for chromatic aberrations, and and spot centring refinement by MMF. Outer kinetochore position was finally corrected for chromatic aberrations by enforcing that the cell-average distance between inner and outer kinetochore markers in each microscope x-, y-, and z-coordinate was equal to zero, as previously demonstrated to be the case after accurate correction for chromatic aberrations[42]. Preliminary chromatic aberrations were calculated daily as the average distance in x, y, and z between Oregon Green- and TMR-labelled in HeLa HaloTag-CENPA cells. Kinetochores were manually paired to generate lists of kinetochore sisters. The intra-kinetochore distance was defined as the distance between the inner and outer kinetochore markers, pointing towards the outer kinetochore. The 3D intra-kinetochore distance was calculated as the 3D Euclidean distance between inner and outer kinetochore markers for each kinetochore. The 3D swivel was defined as the angle tended between the intra-kinetochore axis (vector pointing from the inner to the outer marker) and the inter-kinetochore axis (vector lying through both inner markers for a given sister pair, pointing towards the kinetochore in question; Fig. 2c). The 3D measurements were corrected per experiment for inflationary effects[70] by fitting a maximum likelihood fit to the probability distribution:

$$p_{3D} = \sqrt{\frac{2}{\pi}} \cdot \frac{\Delta_{3D}}{\sigma\mu} \cdot \exp\left(-\frac{\Delta_{3D}^2 + \mu^2}{2\sigma^2}\right) \cdot \sinh\left(\frac{\mu\Delta_{3D}}{\sigma^2}\right),$$

where μ and σ are the "true" mean 3D intra-KT distance and measurement standard deviation, respectively, for a dataset of 3D Euclidean intra-KT measurements, $\Delta_{3D}$. Mean values given in Fig. 2d are the mean inflation-corrected distances from $N = 4$ experiments. Y-swivel was defined as the angle tended between the intra-KT axis (vector pointing from the inner to the outer marker) and the inter-KT axis (vector lying through both inner markers for a given sister pair, pointing towards the KT in question; Fig. 2c) in the xy plane.

**Image processing and analysis of k-fibre intensities**. The 3D confocal images were analysed with a custom-made framework written in MATLAB (The Math-Works, Inc., Natick, MA, USA) version 1.2. Briefly, based on the CENPA channel the user identifies KTs within a focal plane. Based on the brightness of the tubulin channel the code identifies the MT bundle associated to a given KT within 400 nm. To quantify k-fibre intensity the code calculates the mean tubulin intensity in a region of interest of 500 × 420 nm that is centred on the brightest tubulin spot and that is oriented along the axis of the MT bundle. The local background signal, whose position is determined by the user, is subtracted from the obtained intensity.

**Statistical analysis**. All experiments are based on at least three independent experiments. Numbers of independent experiments (N) and number of analysed cells/KTs (n) are indicated in the figure legends. All statistical evaluations were run on PRISM 7.02 (GraphPad, USA); the specific statistical tests and the p values are indicated in the figure legends. Graphs were plotted in PRISM 7.02 and mounted in Adobe Illustrator (Adobe).

**Code availability**. The fully modifiable MATLAB code to quantify the intensity of the KT-fibres is available under https://github.com/Bioimaging/Kinetochore-Microtubule-Intensity.

**Data availability**. The authors declare that all data supporting the findings of this study are available within the article and its supplementary information files or from the corresponding author upon reasonable request. The primary and secondary data generated in the course of this project are available upon request. Due to their large size they can only be sent on external hard disks.

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

## Acknowledgements

The authors are grateful to I. Cheeseman (MIT, USA), J. DeLuca (Colorado State University, USA), J. Pines (The Institute of Cancer Research, London, UK), W. Krek (ETH Zurich, Switzerland), A. Khodjakov (NY State University, USA), D. Gerlich (IMBA, Austria), and H. Lane and F. Bachmann (Basilea Pharmaceutica Ltd, Switzerland) for reagents, the Bioimaging and the Electron Microscopy facilities at the Medical Faculty of the University of Geneva for help in sample preparation, Pierre Cosson (University of Geneva, Switzerland) and Stephen Royle (University of Warwick, UK) for help in analysis of electron microscopy data, Catharina Samora (University of Warwick, UK) for the generation of the PRC1 antibody, and the members of the Meraldi laboratory for helpful discussions. Work in the Meraldi laboratory is supported by the SNF-project grant (No 31003A_160006), a research grant from Basilea Pharmaceutica International Ltd, and the University of Geneva. Work in the McAinsh laboratory is supported by a Wellcome Investigator Award (grant no. 106151/Z/14/Z) and a Royal Society Wolfson Research Merit Award (grant WM150020). The visit of D. Dudka to the McAinsh laboratory was supported by a short-term EMBO fellowship (ASTF 178-2016).

## Author contributions

The project was initiated by A.N. and P.M. and directed by P.M. D.D. and A.N. carried out all the experiments. C.A.S. and A.D.M. carried out and analysed the high-resolution tracking experiments. N.L. wrote the code for the automated *k*-fibre intensity analysis. All authors interpreted the data and D.D. and P.M. wrote the manuscript with input of the other authors.

## Additional information

**Competing interests:** The Meraldi laboratory was receiving a research grant from Basilea Pharmaceutica International when it made the initial observation that BAL27862 affects MT occupancy and inter-KT distance in dividing cells. All remaining authors declare no competing interests.

