## [Peer Review File · Nature Communications]

Reviewers' comments:

Reviewer #1 (Remarks to the Author):

The work of Dudka, Noatynska and colleagues tackles an interesting and important question that has long puzzled cell division researchers. Why do kinetochores in many organisms attach to upwards of 20 MTs? Relatedly, how does the kinetochore (and cell) respond to partial occupancy when kinetochores can be attached and even bioriented, but do not have the full complement of 20 MTs. To address these questions, the authors made use of low doses of the MT destabilizing drug BAL27862 (BAL). Careful analyses are applied to cells treated with BAL demonstrating that the drug reduces KT-MT occupancy leading to significantly reduced inter-, but not intra-kinetochore distances and normal SAC satisfaction. The drug treatment also yields a small, but statistically different increase in AURBK-dependent phosphorylation of outer kinetochore components, which is not sufficient to generate unattached kinetochores as Mad1 and Mps1 levels are lost/reduced at bioriented kinetochores to the same extent as controls. Correction of syntelic attachments appears normal in BAL-treated cells; however, lagging chromosomes, which the authors attribute to merotelic attachments (see comment below), are increased and anaphase velocities are reduced.

I found this manuscript to be clearly written and to employ a thorough range of techniques to address an important unresolved question in cell division. The observation that full intra-kinetochore distance can be achieved in the absence of full occupancy is informative about the structural nature of the kinetochore and the fact that SAC satisfaction did not correlate with reduction in inter-kinetochore distances is also of note to the field. My most significant concerns do not relate to the data itself, but rather the interpretation of some results. If these concerns are addressed than I am enthusiastic about the work being published in Nature Communications.

Major Points:

1) The establishment of reduced occupancy in Figure 1 is critical to the interpretation of all other results. The identification of kinetochore fibers as being in the vicinity (1 μ m) of chromatin is concerning. It is important that kinetochores are identified in serial planes and the bundles that are being quantified are indeed k-fibers with close proximity to the kinetochore, not just the chromatin. In cells with compromised KT-MT attachments we often see long interpolar bundles spanning the length of the spindle and passing in the vicinity of chromatin. The validity of these counting experiments is essential as the paper is about KT-MT occupancy so further validation of reduced occupancy is warranted. Fluorescence measurements of k-fiber intensity could bolster the case.

2) Along these lines, I'd like to see a better characterization of the spindle morphology as the authors simply state that it is "intact". The supplemental figure 1 is somewhat informative, but since it's potentially supportive of the EM data more cells with tubulin staining should be imaged and quantified. The supplemental figure (as well as the quantification of k-fiber signal) should also be moved to the main figures.

3) The authors consistently present inter-kinetochore distance as the legitimate measure of tension as low inter-kinetochore distances are the definition of "no tension" in their figures. This seemingly simple measurement of inter-kinetochore distance is more complex than merely reflecting tension generated by biorientation. Inter-kinetochore distances are a result of many factors - a combination of end-on dynamics, lateral KT-MT interactions, flux rates, KT-associated motor activities, etc. In this regard, a more careful characterization of spindle properties would better contextualize the interpretation of the reduced inter-kinetochore distances. For example, work recently published in Nature Communications identified "bridging fibers" as important contributors to sister kinetochore tension/distances. Are bridging fibers reduced in the BAL-treated cells (PRC1 localizes to these structures)? See more suggestions (flux) below.

4) While I fully respect the authors ability to speculate in the discussion section (which should be strengthened), I am not 100% convinced that force imbalances or lower force generation by k-fibers explains merotelic attachment frequency or reduced anaphase velocities respectively.

5) Regarding anaphase velocities, kinetochores move rapidly poleward during prophase (much, faster than during anaphase) when lower pole-directed forces (by KT-dynein) are applied than the

upper bound of forces that could be generated at a full occupancy KT during anaphase. Also, while the potential to generate very high forces during anaphase exists, it is often proposed that very low force is sufficient to move sister chromatids at anaphase velocities through the cytosol and anaphase force estimates as low as ~ 1 pN have been made. There are numerous possible explanations, aside from the proposed anaphase force generation idea, for a reduction in anaphase velocity. One obvious parameter that comes to mind is flux rates. It would be worthwhile to measure flux rates in the presence of BAL to test this possibility.

6) I am certainly convinced that the BAL treatment leads to an increase in lagging chromosomes, to conclusively demonstrate that they are merotelic requires tubulin visualization (in keeping with the points above). If they are indeed merotelic attachments, the authors' argument is that lower occupancy results in lower force differential between KT-MTS attached to the opposing poles and thus the BAL merotelics are not resolved as quickly as in control conditions. While I buy this idea more than the authors' explanation of the reduced anaphase velocity there are certainly other possibilities for the increase in merotelic that are not discussed. For example, what if doses of the drug reduce KT-MT turnover enough to increase merotelic. I don't feel additional experiments measuring KT-MT turnover are necessary but a discussion of other possibilities would broaden the perspective.

Minor Comment:

1) The Mad2 panel in nocodazole-treated cells in 2J looks funky. The highlighted pair in the Mad1 inset looks to be different than the pair in the Mps1 and CENPA channels.

Reviewer #2 (Remarks to the Author):

In this study, the authors use a new microtubule (MT) targeting agent (BAL27862) that causes reduced MT occupancy at the kinetochore (KT) and reduced inter-KT stretch (tension). They found that reduced occupancy and tension caused slight perturbations of chromosome behavior (e.g., swivel, oscillation period, and chromosome alignment timing), but did not impair overall mitotic progression, mitotic checkpoint function, or Aurora B kinase-dependent phosphorylation of KT substrates (which implies no effect on Aurora B-dependent correction of KT mis-attachments). However, they found a slight increase in anaphase lagging chromosomes in a monastrol washout experiment. They also found that overall anaphase chromosome movement was slower in BAL27862-treated cells. Based on these observations, the authors conclude that full MT occupancy in higher eukaryotes ensures higher size differential between the two MT bundles attached to merotelic KTs. This differential is known to ensure movement of merotelically attached KTs away from the spindle equator. Overall, the work presented here is of high quality and the manuscript is very well written. However, I am not fully convinced by the major conclusion drawn by the author with respect to the mechanism behind the increased rate of anaphase lagging chromosomes. This major issue and some minor issues are listed below and will have to be addressed before publication.

Major issue:

The major issue in my opinion is related to the data presented in figure 4 F-G. The authors measured velocity of anaphase chromosome movement by tracking the distance between sister KTs over time after anaphase onset. However, there are two factors that determine overall anaphase chromosome movement, one being the movement of the individual chromosome toward its respective pole (A.K.A. anaphase A) and the other being the movement of the two spindle poles away from each other (A.K.A. anaphase B). Unfortunately, in most cell types the time difference between beginning of anaphase A and beginning of anaphase B is minimal and the way chromosome movement was tracked in the experiments presented here does not allow the separation of chromosome-to-pole movement from movement due to spindle elongation. In figure

1A, the authors point out that BAL27862 treatment also resulted in a reduction of non-KT MTs. Because non-KT MTs play a role in anaphase B spindle elongation, the reduced velocity of anaphase chromosome movement could be explained by a slower rate of spindle elongation. Importantly, a slower anaphase B rate could also explain the increase in anaphase lagging chromosomes. Indeed, the study the authors cite (Cimini et al., 2004) proposed that movement of merotelic KT away from the spindle equator during anaphase is caused by lengthening of the thinner MT bundle, which in turn is caused by the tension experienced by the MTs as the spindle elongates and produces a force on the merotelically attached KT. Thus, unless the authors can provide evidence that chromosome-to-pole movement, but not spindle elongation, is specifically impaired in the presence of BAL27862, the major conclusion of the manuscript cannot be supported. The key experiment would consist of KT tracking in cells with fluorescent tags at both KTs and spindle poles, so that movement due to anaphase A vs. anaphase B can be separated.

Minor issues:

1. The introduction is a little long.
2. Introduction, line 39. Neither of the papers cited here (references 8 and 9) actually showed that a single MT can move a higher eukaryotic chromosome. So, this statement is mis-leading. Whereas it is true that in budding yeast a single MT is enough to move a chromosome, chromosome size may be a factor.
3. Line 68. The sentence starting with "This question" is confusing as there is no question in the preceding sentence.
4. Line 86. In the sentence starting with "It" it is not clear what "It" refers to.
5. Line 161. For clarity, it would be better to add the word "average" before "inter-kinetochore distances." Moreover, referring to this reduction as "no tension" is quite confusing.
6. Line 192. It is not clear what subpopulation (the subpopulation of 9 or 3 cells?) the authors refer to when they write "this subpopulation."
7. In the section on pages 10-11, it would be better to refer to figure S2 earlier and throughout instead of just at the very end. It is also not clear why Ndc80 phosphorylation was not analyzed in the presence of monastrol as for Dsn1.
8. Line 256. The word "but" should be deleted.
9. Panels C and D of figure 4 are cited before A and B. I'm also not sure whether the schematic in 4A is necessary.
10. Line 272. The word "cells" is missing after "BAL27862-treated"
11. Line 307. I suggest replacing the word "failing" with "lagging"
12. Line 308. Please, replace "instable" with "unstable"
13. Line 328. It is not clear why reference 66 is cited here, when talking about the role of Aurora A in destabilizing mis-attachments on polar chromosomes.
14. Some references are incomplete. The ones I noted include: 12, 28, 30, 36, 51, and 63.
15. It is not clear why the dose of the drug varied throughout the paper between 12 and 15 nM.

16. In several experiments, the sample size for the treated cells was substantially smaller. It would be important for the authors to clarify if this difference was due to a smaller number of cells being in mitosis in the treated population. This could be evaluated by a simple mitotic index and it would be important to know if BAL27862 also had an effect at some other stage of the cell cycle, thus diminishing the number of cells entering mitosis.

17. The font size in the labels of some graphs is very small. I am afraid that it will become too small to read once the figures are re-sized for publication.

18. Although the manuscript is overall very well written, there are issues with punctuation (particularly missing commas) throughout the text.

19. In a number of graphs, the units are incorrect or not present. The ones I noticed include: 1I-K (both X axis and legend), 2E-F (Y axis and labels on graph); 4F (X axis).

20. Figure 1A. The green circles are not easy to see. Yellow may be a better color choice.

21. Figure 1J-K. There is a misspelling in the Y axis labels. It should read "# of occurrences"

22. Figure 2K-L. The label in the graph titles "(in subpopulation with no tension)" is confusing, given that the DMSO sample includes all the cells. Based on figure 1I, there appears to be a small subpopulation of DMSO cells with low tension. It would better to use only that subpopulation in K and L.

23. Figure 1 H and L. It would better to display panel H right above panel L and to resize the two graphs so that they have the same width. This would allow easier comparison of the data.

24. Figure 2F. The Y axis should read "Average inter-....."

25. Figure 3 and S2. I think it would be better to display all the data in one graph, with everything normalized to the DMSO alone. However, I understand if the authors chose to display the data as they did, if the data were collected at different times.

26. Figure 4A. As mentioned earlier, I am not sure the schematic in A is really necessary. If this is kept, however, the authors may choose to change their color scheme, given that the syntelic attachment in the first panel is hard to see.

27. Figure 4G. I believe "velocity" may be a better word choice than speed.

28. Figure 4H, bottom. I have an issue with the "resolve merotelics" terminology here. The merotelic attachment is actually not resolved in anaphase, because MT attachment is maintained. So, the terminology is confusing. I suggest replacing the word "resolve" with "segregate"

Reviewer #3 (Remarks to the Author):

Dudka et al. use a 'new' microtubule drug (BAL27862) to probe the relationship between microtubule attachment, 'tension', spindle assembly checkpoint (SAC) activation, and correction of erroneous kinetochore-microtubule attachments. The authors' ability to create genetically unaltered kinetochores at a steady-state with low numbers of attached microtubules and no centromere stretch represents a technical advance. They use this ability to demonstrate that neither centromere stretch nor a full metaphase complement of microtubules is required to satisfy the SAC. Importantly, while recent work in the field has suggested these conclusions, this study

demonstrates them with unperturbed (or at least undetectably perturbed) kinetochores. Additionally, the findings that a loss of centromere stretch on bioriented kinetochores does not lead to full destabilization of kinetochore-microtubule attachments, and the findings suggesting that a large number of kinetochore-microtubule attachments may be required for resolving merotelic attachments, both represent conceptual advances. Most of these conclusions are well supported. We have only two major comments.

Major points:

1) How BAL27862 affects microtubules is not clear. First, the EM in Figure 1 is not clear and clearer data needs to be presented. Are there any orthogonal ways the authors can show the effect of BAL27862 on microtubule number? The example images with circled density just do not look different from the surrounding density. This is critical in order to be able to interpret the effects of BAL27862. Second, information on how the drug affects k-fiber microtubule stability (e.g. cold stability) and dynamics (e.g. poleward flux) should be presented. Just as microtubule numbers may be important for function, microtubule stability/dynamics may also be important for function and may impact merotelic attachment resolution. It is important to determine all effects the drug may have (or least reasonably attempt to do so) before concluding that a change in microtubule number leads to less efficient merotelic resolution. Third, information on how the drug affects non-k-fiber microtubules in the spindle should be presented, and the authors should decide what aspects of these microtubules are most important to report. This is interesting in its own right, and important to interpret the effects of the drug on spindle function. The above seem at least as – and if not more – important than some of the details (intra-k stretch, swivel, etc.) currently presented in Figure 1.

2) Language throughout the paper regarding “tension” is unclear (e.g. lines 157-159). It would be much clearer to replace “tension” with wording such as “centromere stretch” or “inter-kinetochore stretch”. How is it possible that there can be no tension and still have kinetochore oscillations relatively unchanged? This is unclear and further, it would be worth commenting on the relationship between microtubule number and force generation at metaphase. To this end, instead of just reporting sister kinetochore autocorrelation, it would be useful to report whether metaphase kinetochore velocities are different (since they do see a difference for anaphase velocities). Finally, the authors need to write the anaphase section more clearly. Their current arguments assume that force and velocity are proportional, and I do not believe that a force-velocity relationship is known (e.g. it is possible that velocity does not change with force in a given regime). If I am wrong, the authors should cite papers to back up their assumption and current language use.

Minor points

1. The swivel measurements in Figure 1C-G, while interesting, have standard deviations high enough that it is not clear whether there is sufficient resolution to rule out a difference in kinetochore structure when BAL27862 is added. We recommend removing these panels.
2. Figure 1I and K: need to clarify how they selected the subpopulation (e.g. plot title Figure 1K is unclear). How does the variation in inter-kinetochore distance within a cell compare to the variation between cells?
3. In Figure 2 it is clear that cells treated with BAL27862 are able to satisfy the SAC and completely remove Mps1 from kinetochores, however the conclusion stated in the associated text (lines 216-229 and lines 313-315) is not fully supported by the data and should be softened. Complete Mps1 removal could suggest that the same number of Ndc80 bind to fewer microtubules in the BAL27862 treated cells, or that not every Ndc80 is saturated with Mps1 and it doesn't matter for every Ndc80 to bind to microtubules. It could also be the case that changes in kinetochore phosphorylation state brought on by loss of Mps1 on some number of Ndc80 complex could lead to the loss of Mps1 from the unattached Ndc80 that remain.
4. Figure 2E needs units on the X-axis and a clearer description in the figure legend of what corresponds to time zero.

5. Across the whole paper, need to use "µm" to indicate microns.
6. Figure 4E, is a paired t-test actually appropriate here? Which data are paired?
7. For the anaphase speed reduction (Figure 4G), do they include cells with merotelic kinetochores? Or do they exclude those kinetochores in those cells? It would help to provide more information in the methods about which kinetochores were included in this dataset.
8. The last sentence in the intro mentions "... for the first time ...". This is not needed and likely not completely true. In this regard, the authors should refer to the MUG O'Connell/Khodjakov JCB 2008 paper somewhere in their work.

Point by point response

Reviewer 1

The work of Dudka, Noatynska and colleagues tackles an interesting and important question that has long puzzled cell division researchers. Why do kinetochores in many organisms attach to upwards of 20 MTs? Relatedly, how does the kinetochore (and cell) respond to partial occupancy when kinetochores can be attached and even bioriented, but do not have the full complement of 20 MTs. To address these questions, the authors made use of low doses of the MT destabilizing drug BAL27862 (BAL). Careful analyses are applied to cells treated with BAL demonstrating that the drug reduces KT-MT occupancy leading to significantly reduced inter-, but not intra-kinetochore distances and normal SAC satisfaction. The drug treatment also yields a small, but statistically different increase in AURBK-dependent phosphorylation of outer kinetochore components, which is not sufficient to generate unattached kinetochores as Mad1 and Mps1 levels are lost/reduced at bioriented kinetochores to the same extent as controls. Correction of syntelic attachments appears normal in BAL-treated cells; however, lagging chromosomes, which the authors attribute to merotelic attachments (see comment below), are increased and anaphase velocities are reduced.

I found this manuscript to be clearly written and to employ a thorough range of techniques to address an important unresolved question in cell division. The observation that full intra-kinetochore distance can be achieved in the absence of full occupancy is informative about the structural nature of the kinetochore and the fact that SAC satisfaction did not correlate with reduction in inter-kinetochore distances is also of note to the field. My most significant concerns do not relate to the data itself, but rather the interpretation of some results. If these concerns are addressed than I am enthusiastic about the work being published in Nature Communications.

We are glad that the reviewer 1 is enthusiastic about our work and considers it important for the field. We addressed his/her concerns as follows:

Major comments:

1) The establishment of reduced occupancy in Figure 1 is critical to the interpretation of all other results. The identification of kinetochore fibers as being in the vicinity (1 μ m) of chromatin is concerning. It is important that kinetochores are identified in serial planes and the bundles that are being quantified are indeed k-fibers with close proximity to the kinetochore, not just the chromatin. In cells with compromised KT-MT attachments we often see long interpolar bundles spanning the length of the spindle and passing in the vicinity of chromatin. The validity of these counting experiments is essential as the paper is about KT-MT occupancy so further validation of reduced occupancy is warranted. Fluorescence measurements of k-fiber intensity could bolster the case.

We agree with reviewer 1 that an additional validation of microtubule occupancy strengthens the manuscript. However, due to the need to enrich for mitotic cells (mitotic shake off followed by mild centrifugation on a EM-grid), the spindles of our mitotic cells are oriented randomly. Therefore, obtaining enough serial cuts that are perpendicular to k-fibres for 3D transmission electron microscopy reconstructions, and at a sample size that is sufficient for meaningful statistics, is beyond the scope of a revision, as it would take years. We tried to use a Scanning Electron Microscope / Focused Ion Beam (SEM/FIB) to record 3D EM scans, but microtubules

stained poorly, and SEM/FIB microscopy only offers a 5 nm resolution, compared to the 1 nm resolution of classical transmission electron microscopy, which is problematic when searching for 22 nm wide microtubules. These challenges were the reason we used the standard established by Steve Royle (Booth et al., 2011, EMBO J) to reliably identify k-fibres. Nevertheless, as suggested by reviewer 1 we validated our findings by immunofluorescence, using an hTert-RPE1 cell line stably expressing GFP- α -tubulin. We FACS-sorted the cell line to ensure a homogenous GFP- α -tubulin expression, before combining a fixation method established by the DeLuca laboratory (Zaytsev et al., 2014, JCB) and an in-house MATLAB-based code for semi-automatic quantification of k-fibres intensities (provided with the manuscript). When we quantified the intensities of over 1000 k-fibres per condition, we found a 35% decrease in kinetochore-fibre intensity in BAL27862-treated cells, perfectly corroborating our electron microscopy results (new Fig. 1c and d).

2) Along these lines, I'd like to see a better characterization of the spindle morphology as the authors simply state that it is "intact". The supplemental figure 1 is somewhat informative, but since it's potentially supportive of the EM data more cells with tubulin staining should be imaged and quantified. The supplemental figure (as well as the quantification of k-fiber signal) should also be moved to the main figures.

To better characterize the mitotic spindle in BAL27862-treated cells we performed the following additional experiments:

- 1) In Figure 1e we present a panel of 10 DMSO or BAL27862-treated cells that were fixed under conditions that best preserve kinetochore-microtubules (Zaytsev et al., 2014, JCB), to let the reviewer better evaluate the morphology of the spindle. This panel illustrates the reduced intensity of k-fibres, yet shows that the general morphology of the spindle is preserved.
- 2) In Figure 1f we also quantified spindle length, and show that nanomolar doses of BAL27862 results in a minimal reduction in spindle length.
- 3) In Supplementary Figure 2, we show that nanomolar doses of BAL27862 lead to small astral microtubule asters, consistent with previous studies (Prota et al., 2014, J. Mol. Biol).
- 4) Finally, using PRC1 staining as a marker, we show in Figure 6e and f that BAL27826 treatment leads to a mild (20%) reduction in interpolar microtubules.

Overall, these results show that although nanomolar concentrations of BAL27862 mildly affect all the subsets of spindle microtubules, it does not cause major defects in the overall spindle morphology, nor does it majorly affect chromosome alignment or the timing of anaphase onset (Figure 3).

3) The authors consistently present inter-kinetochore distance as the legitimate measure of tension as low inter-kinetochore distances are the definition of "no tension" in their figures. This seemingly simple measurement of inter-kinetochore distance is more complex than merely reflecting tension generated by biorientation. Inter-kinetochore distances are a result of many factors - a combination of end-on dynamics, lateral KT-MT interactions, flux rates, KT-associated motor activities, etc. In this regard, a more careful characterization of spindle properties would better contextualize the interpretation of the reduced inter-kinetochore distances. For example, work recently published in Nature Communications identified "bridging fibers" as important contributors to sister kinetochore tension/distances. Are bridging

fibers reduced in the BAL-treated cells (PRC1 localizes to these structures)? See more suggestions (flux) below.

We do not believe that the additional factors mentioned by reviewer 1 are important for the inter-kinetochore distance once sister-kinetochores are attached in a bipolar manner. KT-associated motor activities and lateral KT-MT attachments affect inter-kinetochore distances during congression, but once bi-orientation is achieved, it is not thought that they affect inter-kinetochore stretching. Consistently, inter-kinetochore stretching can be fully suppressed by high doses of taxol, which indicates that it mainly depends on microtubule dynamics. This also excludes a priori a contribution of bridging fibres, which depend on motor-driven anti-parallel microtubule sliding (Vukusic et al., 2017, Dev. Cell).

With regard to flux, it has been shown to influence inter-kinetochore distances in *Xenopus* egg extracts (Maddox et al., 2003, JCB), but in human mitotic cells the case is less clear: Ganem and colleagues (2005, Curr. Biol) found that MCAK/Kif2a depletion, which amongst other things abrogates flux, led to a minimal (10%) reduction in inter-kinetochore distances; moreover, CENP-E depletion, which reduces the flux rate by 50% (Maffini et al., 2009, Curr. Biol) does not change inter-kinetochore distances of bipolarly attached sister-kinetochores (Jaqaman et al., 2010, JCB). Therefore, we feel confident that low inter-kinetochore distances in BAL27862-treated cells result majorly from low microtubule occupancy.

Nevertheless, since both bridging fibres and flux are involved in chromosome segregation, we measured both parameters in BAL27862 treated cells and discuss them in the context of anaphase forces. We quantified the amount of bridging fibres with PRC1 antibodies and found a 24.5% reduction (new Fig. 6e and f; see more specifically point 1 of reviewer 2), and used photo-activation to show that BAL27862 treatment led to a 37% decrease in flux rates (see point 5 further below).

4. While I fully respect the author's ability to speculate in the discussion section (which should be strengthened), I am not 100% convinced that force imbalances or lower force generation by k-fibres explains merotelic attachment frequency or reduced anaphase velocities respectively.

We appreciate the concerns of reviewer 1, and we aimed to give a more balanced view in our discussion, emphasizing that microtubule occupancy is most likely the important, but not the only parameter that influences the incidence of lagging chromosomes. With respect to his/her specific concerns, we addressed them under point 5 and 6.

5) Regarding anaphase velocities, kinetochores move rapidly poleward during prophase (much, faster than during anaphase) when lower pole-directed forces (by KT-dynein) are applied than the upper bound of forces that could be generated at a full occupancy KT during anaphase. Also, while the potential to generate very high forces during anaphase exists, it is often proposed that very low force is sufficient to move sister chromatids at anaphase velocities through the cytosol and anaphase force estimates as low as ~1 pN have been made. There are numerous possible explanations, aside from the proposed anaphase force generation idea, for a reduction in anaphase velocity. One obvious parameter that comes to mind is flux rates. it would be worthwhile to measure flux rates in the presence of BAL to test this possibility.

We thank the reviewer 1 for suggesting an alternative hypothesis to the decreased anaphase velocities that we observe upon BAL27862 treatment. First, we excluded any effect originating

from pole-to-pole separation (anaphase B; see point 1 of reviewer 2). Second, as suggested we compared poleward microtubule flux in DMSO and BAL27862-treated cells to find a reduction of 0.4 $\mu\text{m}/\text{min}$ (new Figure 6i and j). Since anaphase A speed is lowered in total by 1.5 $\mu\text{m}/\text{min}$ (Figure 6d), we conclude that the reduction in anaphase A speed in BAL27862-treated cells is most likely a combination of two different factors: a lower microtubule occupancy, which decreases the pulling forces on sister-kinetochores and the corresponding anaphase A velocity by 1.1 $\mu\text{m}/\text{min}$, and a reduction in minus-end depolymerization due to a slower poleward microtubule flux that diminishes anaphase A velocity by 0.4 $\mu\text{m}/\text{min}$. These novel findings allowed us to obtain a more balanced view on anaphase forces, which we discuss in the revised manuscript.

6) I am certainly convinced that the BAL treatment leads to an increase in lagging chromosomes, to conclusively demonstrate that they are merotelic requires tubulin visualization (in keeping with the points above). If they are indeed merotelic attachments, the authors argument is that lower occupancy results in lower force differential between KT-MTS attached to the opposing poles and thus the BAL merotelics are not resolved as quickly as in control conditions. While I buy this idea more than the authors' explanation of the reduced anaphase velocity there are certainly other possibilities for the increase in merotelically attached kinetochores that are not discussed. For example, what if doses of the drug reduces KT-MT turnover enough to increase merotelically attached kinetochores. I don't feel additional experiments measuring KT-MT turnover are necessary but a discussion of other possibilities would broaden the perspective.

To address the first concern we now include in Figure 5e a large panel of immunofluorescence pictures of BAL27862-treated cells that were released from a monastrol block and that contain lagging chromosomes. Due to high microtubule density within the spindle, this approach does not always allow to unequivocally determine the status of kinetochore-microtubule attachments. Nevertheless, we easily found multiple cases of unresolved merotelically attached kinetochores, implying that the higher incidence of lagging chromosomes indeed originates from merotelic attachments.

Concerning the second point, a reduced KT-MT turnover could indeed increase the frequency of merotelic attachments due to an impaired error-correction. To address this potential caveat we determined microtubule stability after a cold treatment, which reflects KT-MT turnover. We found that low nanomolar doses of BAL27862 leads to small but reproducible decrease in KT-MT stability, implying a higher KT-MT turnover (new Figure 6a and b). We conclude that the higher incidence of kinetochore-microtubule does not originate from a lower KT-MT turnover, a result that further supports our model where a strong force differential helps to mechanically segregate merotelic attachments.

Minor Comments:

1. The Mad2 panel in nocodazole-treated cells in 2J looks funky. The highlighted pair in the Mad1 inset looks to be different than the pair in the Mps1 and CENPA channels.

We thank the reviewer 1 for spotting this mistake. Indeed the insert presented in the Fig. 2j (now Fig. 3j) was indicating a wrong sister-kinetochore pair. We corrected it.

Reviewer 2

In this study, the authors use a new microtubule (MT) targeting agent (BAL27862) that causes reduced MT occupancy at the kinetochore (KT) and reduced inter-KT stretch (tension). They found that reduced occupancy and tension caused slight perturbations of chromosome behavior (e.g., swivel, oscillation period, and chromosome alignment timing), but did not impair overall mitotic progression, mitotic checkpoint function, or Aurora B kinase-dependent phosphorylation of KT substrates (which implies no effect on Aurora B-dependent correction of KT mis-attachments). However, they found a slight increase in anaphase lagging chromosomes in a monastrol washout experiment. They also found that overall anaphase chromosome movement was slower in BAL27862-treated cells. Based on these observations, the authors conclude that full MT occupancy in higher eukaryotes ensures higher size differential between the two MT bundles attached to merotelic KTs. This differential is known to ensure movement of merotelically attached KTs away from the spindle equator. Overall, the work presented here is of high quality and the manuscript is very well written. However, I am not fully convinced by the major conclusion drawn by the author with respect to the mechanism behind the increased rate of anaphase lagging chromosomes. This major issue and some minor issues are listed below and will have to be addressed before publication.

We thank the reviewer 2 for appreciating the quality of both our work and the manuscript. We addressed his/her major and minor comments in the following manner:

Major comments:

1. The major issue in my opinion is related to the data presented in figure 4 F-G. The authors measured velocity of anaphase chromosome movement by tracking the distance between sister KTs over time after anaphase onset. However, there are two factors that determine overall anaphase chromosome movement, one being the movement of the individual chromosome toward its respective pole (A.K.A. anaphase A) and the other being the movement of the two spindle poles away from each other (A.K.A. anaphase B). Unfortunately, in most cell types the time difference between beginning of anaphase A and beginning of anaphase B is minimal and the way chromosome movement was tracked in the experiments presented here does not allow the separation of chromosome-to-pole movement from movement due to spindle elongation. In figure 1A, the authors point out that BAL27862 treatment also resulted in a reduction of non-KT MTs. Because non-KT MTs play a role in anaphase B spindle elongation, the reduced velocity of anaphase chromosome movement could be explained by a slower rate of spindle elongation. Importantly, a slower anaphase B rate could also explain the increase in anaphase lagging chromosomes. Indeed, the study the authors cite (Cimini et al., 2004) proposed that movement of merotelic KTs away from the spindle equator during anaphase is caused by lengthening of the thinner MT bundle, which in turn is caused by the tension experienced by the MTs as the spindle elongates and produces a force on the merotelically attached KT. Thus, unless the authors can provide evidence that chromosome-to-pole movement, but not spindle elongation, is specifically impaired in the presence of BAL27862, the major conclusion of the manuscript cannot be supported. The key experiment would consist of KT tracking in cells with fluorescent tags at both KTs and spindle poles, so that movement due to anaphase A vs. anaphase B can be separated.

To address the reviewer's concerns we now first investigated how nanomolar doses of BAL27862 might affect astral microtubules and bridging fibres, which are both potentially involved in spindle elongation (new Supplementary Figure S2 and Figure 6e and f), and measured spindle elongation itself (new Figure 6g and h). Although we found a minor reduction for astral microtubule and

bridging fibre intensities, when we quantified the speed of anaphase B in an hTert-RPE1 GFP-CENPA (kinetochore marker) and Centrin1-GFP (centrosome marker) cell line, we found no differences. We conclude that a BAL27862-treatment does not affect the rate of spindle elongation. This important complementary experiment therefore corroborates our conclusion that the persistence of merotelic attachments is caused by an insufficient kinetochore-microtubule driven anaphase A force.

Minor comments:

1. The introduction is a little long.

As suggested by the reviewer, we somewhat shortened the introduction, but within limits, since our experiments touch on multiple mitotic processes, such as force generation at kinetochore, spindle assembly checkpoint satisfaction, error-correction of syntelic and merotelic kinetochore-microtubule attachments as well as the role of the Aurora-B gradient in these processes. We feel that these elements have to be present in the introduction to give the reader the proper context for our results.

2. Introduction, line 39. Neither of the papers cited here (references 8 and 9) actually showed that a single MT can move a higher eukaryotic chromosome. So, this statement is mis-leading. Whereas it is true that in budding yeast a single MT is enough to move a chromosome, chromosome size may be a factor.

We agree that the cited papers do not fully back up our claim, and that a mis-understanding can arise. However, we emphasize that our original claim was that a single microtubule could generate **sufficient force** to move a higher eukaryotic chromosome. We note that Nicklas and Taylor showed independently that 0.1pN was sufficient to move a vertebrate chromosome (Taylor, 1965; Nicklas, 1965, JCB), and that the McIntosh laboratory showed that single depolymerizing microtubule can generate up to 30pN when efficiently coupled to a kinetochore protein (Volkhov et al, 2013, PNAS). To avoid a mis-understanding we now specifically mention these numbers and citations in the text.

3. Line 68. The sentence starting with “This question” is confusing as there is no question in the preceding sentence.

We thank the reviewer and we corrected this sentence by starting with “This is....”

4. Line 86. In the sentence starting with “It” it is not clear what “It” refers to.

“It” refers to the model described in the previous sentence. We replaced “It is supported by the fact..” by “Consistently...”

5. Line 161. For clarity, it would be better to add the word “average” before “inter-kinetochore distances.” Moreover, referring to this reduction as “no tension” is quite confusing.

We added the word average; moreover to be consistent, we now avoid whenever possible the word tension and rather use the words inter-kinetochore stretching or inter-kinetochore distances (see point 4 of reviewer 3).

6. Line 192. It is not clear what subpopulation (the subpopulation of 9 or 3 cells?) the authors refer to when they write “this subpopulation.”

This was grammatical error, it should say: “These subpopulations” (of 9 and 3 cells).

7. In the section on pages 10-11, it would be better to refer to figure S2 earlier and throughout instead of just at the very end.

We agree and have corrected the mentioned sections accordingly (Note that this is now Figure S1).

It is also not clear why Ndc80 phosphorylation was not analyzed in the presence of monastrol as for Dsn1.

The reasoning behind our experiment was to first prove that BAL27862 does not weaken Aurora B activity. For this we chose phospho-Dsn1 antibodies, which in our hands gave a much better signal than the phospho-Ndc80 antibodies. Moreover, the Ser 100 residue of Dsn1 gives a more sensitive readout of Aurora-B activity as it is localized closer to the centromeric Aurora-B gradient (Wan et al., 2009, Cell). Once this was proven, we used the phospho-Ndc80 antibody to corroborate the phospho-Dsn1 results showing that Aurora-B activity in BAL27862-treated cells is low on aligned sister-kinetochores with a low inter-kinetochore stretch.

8. Line 256. The word “but” should be deleted.

We agree with the reviewer, and have modified the text accordingly.

9. Panels C and D of figure 4 are cited before A and B. I’m also not sure whether the schematic in 4A is necessary.

We thank the reviewer for pointing at wrong order of citations – we now have corrected this paragraph. We also deleted the schematic in Fig. 4a.

10. Line 272. The word “cells” is missing after “BAL27862-treated”

We thank the reviewer for pointing this out.

11. Line 307. I suggest replacing the word “failing” with “lagging”

As we extensively re-wrote the discussion, this sentence has now disappeared. We were nevertheless careful to use a precise vocabulary, such as lagging chromosomes.

12. Line 308. Please, replace “instable” with “unstable”

We corrected this mistake.

13. Line 328. It is not clear why reference 66 is cited here, when talking about the role of Aurora A in destabilizing mis-attachments on polar chromosomes.

In this paragraph we discuss a possible role of Aurora-A in correction of syntelic attachments. We cite Barisic et al, NCB, 2014 because to our knowledge it is the first publication that shows that

Aurora-A inhibition stabilizes end-on attached polar chromosomes (see Figure 4 of Barisic et al, and corresponding discussion). This model was later substantially validated by the Chmatal et al, 2105 and Ye et al, 2015 both Curr. Biol. We feel that nevertheless the work of the Maiato group should be cited in this context.

14. Some references are incomplete. The ones I noted include: 12, 28, 30, 36, 51, and 63.

We thank the reviewer for spotting this. The overlooked errors came from the automatic reference tool we used. This has now been corrected.

15. It is not clear why the dose of the drug varied throughout the paper between 12 and 15 nM.

We found that the effects of BAL27862 show slight variations within different hTert-RPE1 cell lines (our unpublished data). To obtain consistent results (low inter-kinetochore distances and no substantial delay in anaphase onset), we had to adjust the concentrations from cell line to cell line in a range of 12-15 nM. We now explicitly state this in the Material and Methods

16. In several experiments, the sample size for the treated cells was substantially smaller. It would be important for the authors to clarify if this difference was due to a smaller number of cells being in mitosis in the treated population. This could be evaluated by a simple mitotic index and it would be important to know if BAL27862 also had an effect at some other stage of the cell cycle, thus diminishing the number of cells entering mitosis.

We thank the reviewer for this suggestion. However, based on the re-analysis of our live-cell imaging movies, we find no evidence that BAL27862 changes the number of mitotic cells. Both in our overnight movies and after a 4-hour monastrol-treatment with find equivalent number of mitotic cells in DMSO- and BAL27862-treated cells.

17. The font size in the labels of some graphs is very small. I am afraid that it will become too small to read once the figures are re-sized for publication.

We thank the reviewer for helping as make our manuscript easier to read. The small labels have been re-sized and are now compliant with the *Nature Communications* guidelines.

18. Although the manuscript is overall very well written, there are issues with punctuation (particularly missing commas) throughout the text.

We revised the text and tried our best to put back the missing commas.

19. In a number of graphs, the units are incorrect or not present. The ones I noticed include: 11-K (both X axis and legend), 2E-F (Y axis and labels on graph); 4F (X axis).

The “μ” symbol has not been exported correctly from the “adobe illustrator format” to “pdf” format. We corrected it.

20. Figure 1A. The green circles are not easy to see. Yellow may be a better color choice.

The reviewer 2 is right, yellow colour proved to be much more visible than green. We thank the reviewer for this comment.

21. Figure 1J-K. There is a misspelling in the Y axis labels. It should read “# of occurrences”

We corrected this mistake.

22. Figure 2K-L. The label in the graph titles “(in subpopulation with no tension)” is confusing, given that the DMSO sample includes all the cells. Based on figure 1I, there appears to be a small subpopulation of DMSO cells with low tension. It would better to use only that subpopulation in K and L.

We believe the reviewer 2 is referring to the original figures 1k and l. It is true that in both graphs we are using the same DMSO-treated cell population as negative control. This is because there is no subpopulation of DMSO-treated cells with an average low inter-kinetochore distance. The distribution of DMSO-treated cells shows all sister-kinetochores at all time points. Since sister-kinetochores are permanently “breathing”, there are always a number of sister-kinetochores with a low stretching, but on average none of those cells have a low inter-kinetochore distance. Nevertheless, to make this point clearer we now state in the figure that the subpopulation with low sister-kinetochore distances refers only to BAL27862-treated cells.

23. Figure 1 H and L. It would better to display panel H right above panel L and to resize the two graphs so that they have the same width. This would allow easier comparison of the data.

Due to the flow of the arguments, it is unfortunately not possible to place these two graphs above each other. Nevertheless, since in both graphs we use the same standard (DMSO-treated cells) we believe that a meaningful comparison is possible.

24. Figure 2F. The Y axis should read “Average inter-.....”

The reviewer is right. We corrected the axis.

25. Figure 3 and S2. I think it would be better to display all the data in one graph, with everything normalized to the DMSO alone. However, I understand if the authors chose to display the data as they did, if the data were collected at different times.

The data were indeed collected at different times. We thank the reviewer for his/her understanding.

26. Figure 4A. As mentioned earlier, I am not sure the schematic in A is really necessary. If this is kept, however, the authors may choose to change their color scheme, given that the syntelic attachment in the first panel is hard to see.

We agree with the reviewer and removed the scheme.

27. Figure 4G. I believe “velocity” may be a better word choice than speed.

We agree with the reviewer and changed the label accordingly.

28. Figure 4H, bottom. I have an issue with the “resolve merotelics” terminology here. The merotelic attachment is actually not resolved in anaphase, because MT attachment is

maintained. So, the terminology is confusing. I suggest replacing the word “resolve” with “segregate”

We agree with the reviewer that segregating merotelic chromosomes is more accurate, in line with the original description by D. Cimini.

Reviewer 3

Dudka et al. use a ‘new’ microtubule drug (BAL27862) to probe the relationship between microtubule attachment, ‘tension’, spindle assembly checkpoint (SAC) activation, and correction of erroneous kinetochore-microtubule attachments. The authors’ ability to create genetically unaltered kinetochores at a steady-state with low numbers of attached microtubules and no centromere stretch represents a technical advance. They use this ability to demonstrate that neither centromere stretch nor a full metaphase complement of microtubules is required to satisfy the SAC. Importantly, while recent work in the field has suggested these conclusions, this study demonstrates them with unperturbed (or at least undetectably perturbed) kinetochores. Additionally, the findings that a loss of centromere stretch on bioriented kinetochores does not lead to full destabilization of kinetochore-microtubule attachments, and the findings suggesting that a large number of kinetochore-microtubule attachments may be required for resolving merotelic attachments, both represent conceptual advances. Most of these conclusions are well supported. We have only two major comments.

We thank the reviewer 3 for acknowledging that our work provides both technical and conceptual advance for the field. Here we addressed the major and minor comments of the reviewer:

Major comments:

1) How BAL27862 affects microtubules is not clear. First, the EM in Figure 1 is not clear and clearer data needs to be presented. Are there any orthogonal ways the authors can show the effect of BAL27862 on microtubule number? The example images with circled density just do not look different from the surrounding density. This is critical in order to be able to interpret the effects of BAL27862. Second, information on how the drug affects k-fiber microtubule stability (e.g. cold stability) and dynamics (e.g. poleward flux) should be presented. Just as microtubule numbers may be important for function, microtubule stability/dynamics may also be important for function and may impact merotelic attachment resolution. It is important to determine all effects the drug may have (or least reasonably attempt to do so) before concluding that a change in microtubule number leads to less efficient merotelic resolution. Third, information on how the drug affects non-k-fiber microtubules in the spindle should be presented, and the authors should decide what aspects of these microtubules are most important to report. This is interesting in its own right, and important to interpret the effects of the drug on spindle function. The above seem at least as – and if not more – important than some of the details (intra-k stretch, swivel, etc.) currently presented in Figure 1.

First, to improve the resolution of the EM pictures we now zoomed in more on the individual kinetochore-fibres, allowing the reader to better see the typical microtubule rings. We also consulted with Stephen Royle (Univ. of Warwick) an expert in the field, who confirmed that our electron microscopy images were of a good quality and allow counting the number of microtubules in k-fibres (personal communication). Finally, to validate our finding we also

quantified the intensity of k-fibers by immunofluorescence (see point 1 of reviewer 1), and confirmed that BAL27862-treatment reduces k-fibre intensity by 35%.

Second, to address the reviewer's concern on k-fibre cold-stability and poleward microtubule flux we now quantified both parameters (see points 2 and 3 of reviewer 1).

Finally, we regard to the non-kinetochore-microtubule populations we now also monitored the effects of BAL27862 on astral microtubules and bridging fibres (see response to comment 2 of reviewer 1).

2) Language throughout the paper regarding “tension” is unclear (e.g. lines 157-159). It would be much clearer to replace “tension” with wording such as “centromere stretch” or “inter-kinetochore stretch”. How is it possible that there can be no tension and still have kinetochore oscillations relatively unchanged? This is unclear and further, it would be worth commenting on the relationship between microtubule number and force generation at metaphase. To this end, instead of just reporting sister kinetochore autocorrelation, it would be useful to report whether metaphase kinetochore velocities are different (since they do see a difference for anaphase velocities). Finally, the authors need to write the anaphase section more clearly. Their current arguments assume that force and velocity are proportional, and I do not believe that a force-velocity relationship is known (e.g. it is possible that velocity does not change with force in a given regime). If I am wrong, the authors should cite papers to back up their assumption and current language use.

First, we agree that the term “tension” can be vague. We therefore whenever possible use in the revised manuscript the term inter-kinetochore stretching or inter-kinetochore distances, which are more precise. The only exception can be found in the introduction when we discuss the fact that “tension” stabilizes kinetochore-microtubules, and of course in the title of numerous references.

Second, as suggested by the reviewer we now also include in the new Figure 2i the average kinetochore velocity in DMSO and BAL27862-treated cells, showing a significant decrease. This fits with our interpretation that a reduction in kinetochore-microtubule-occupancy reduces the forces acting on kinetochores.

Third, with regard to kinetochore oscillations and inter-kinetochore stretching, we now mention in our discussion that our results are consistent with recent “reverse-engineering” study, based on wild-type kinetochore trajectories, which found that amongst the different forces acting on kinetochores - the kinetochore-microtubule themselves, the polar ejection force, and the spring force that arise from centromere stretching - that the spring force is the weakest force of all (Armond et al., 2015 Plos Comp Biol). This would explain why a severe reduction in this spring force does not impair oscillations, which appear to be mostly driven by kinetochore-microtubule dynamics.

Finally, on the relationship between force and velocity we respectfully disagree. As highlighted by the excellent review on anaphase forces of the Scholey laboratory (Civelekoglu-Scholey and Scholey, 2010 CMLS) forces and velocities are known to be proportional at the sub-cellular level:

“In the sub-cellular regime, low Reynolds number conditions dominate motility, so that inertial forces (due to mass) are negligible in driving movements, and viscous forces (due to friction) have the major effect....

This low value for the Reynolds number means that the chromosomes' movement is dominated by viscous forces, and hence the net force driving the movement of the chromosome is proportional to its velocity and its viscous drag coefficient: $F = lv$

Nevertheless, to help the reader we now explicitly point to this fact.

Minor comments

1. The swivel measurements in Figure 1C-G, while interesting, have standard deviations high enough that it is not clear whether there is sufficient resolution to rule out a difference in kinetochore structure when BAL27862 is added. We recommend removing these panels.

Based on an improvement of the kinetochore-tracking code (available under <https://github.com/cmcb-warwick>), we have re-analysed our recordings, and obtained a much larger sample size (close to 2000 kinetochore-pairs). This shows that if at all, the swivel is lower in BAL27862-treated cells than in DMSO-treated cells (new Figure 2e). We therefore would like to keep this data, as it confirms that BAL27862-treatment does not prevent the formation of bipolar kinetochore-microtubule attachments. Moreover, based on recent publication (Suzuki et al., 2018, eLife), we have now applied a correction to our Euclidian distances for the calculation of intra-kinetochore distances (Figure 2d).

2. Figure 1I and K: need to clarify how they selected the subpopulation (e.g. plot title Figure 1K is unclear). How does the variation in inter-kinetochore distance within a cell compare to the variation between cells?

The variation of inter-kinetochore distances within a cell is higher than the variation amongst the cells, since sister-kinetochores are permanently “breathing”. As we mention in the discussion, this means that at given time point several kinetochores in a cell have inter-kinetochore distances at the resting distance. Therefore, for our analysis we isolated the subpopulation of BAL27862-treated cells with an **average** inter-kinetochore distance of 0.8µm or less and plotted only these cells alongside all the DMSO-treated cells (which on average never have an inter-kinetochore distance of 0.8µm) and all the nocodazole-treated cells (which always have very low inter-kinetochore distances). We modified the text and the figure legend to make this point more clear.

3. In Figure 2 it is clear that cells treated with BAL27862 are able to satisfy the SAC and completely remove Mps1 from kinetochores, however the conclusion stated in the associated text (lines 216-229 and lines 313-315) is not fully supported by the data and should be softened. Complete Mps1 removal could suggest that the same number of Ndc80 bind to fewer microtubules in the BAL27862 treated cells, or that not every Ndc80 is saturated with Mps1 and it doesn't matter for every Ndc80 to bind to microtubules. It could also be the case that changes in kinetochore phosphorylation state brought on by loss of Mps1 on some number of Ndc80 complex could lead to the loss of Mps1 from the unattached Ndc80 that remain.

As suggested by the reviewer we have now softened our conclusion, stating that the complete loss of Mps1 under conditions of partial microtubule-occupancy generally points to a cooperative model. We then in a second step refer to the “Ndc80” lawn model first proposed by (Zaytsev et al., 2014, JCB) as an attractive **possibility** for such a cooperative mechanism, as to our knowledge this is the only model for which there is concrete experimental support. This does, however, not

exclude other models, and our conclusions just state that our result is consistent with the Ndc80 lawn model.

4. Figure 2E needs units on the X-axis and a clearer description in the figure legend of what corresponds to time zero.

The original time zero was arbitrary, since these movies were started in metaphase to record inter-kinetochore distances as cells progressed to anaphase. For the sake of clarity, we now defined in Figure 3e anaphase onset as $t = 0$, and changed the units to minutes. This is of course not possible for the nocodazole-treated cell, which serves as an illustration for a metaphase cell with minimal inter-kinetochore distances.

5. Across the whole paper, need to use “ μm ” to indicate microns.

See minor comment 19 of reviewer 2

6. Figure 4E, is a paired t-test actually appropriate here? Which data are paired?

The data presented in this Figure (new Fig. 5d) is based on 9 independent experiments, in which we each time monitored in parallel DMSO- and BAL27862-treated cells released from a monastrol block (using Ibidi multi-well chambers). Therefore each “paired” set of cells shared the same experimental conditions (temperature, light exposure, monastrol concentrations, the time of release), which may exhibit minor day-to-day variations and which may have a minor effect of the incidence of chromosome segregation errors. A paired t-test is therefore adequate.

7. For the anaphase speed reduction (Figure 4G), do they include cells with merotelic kinetochores? Or do they exclude those kinetochores in those cells? It would help to provide more information in the methods about which kinetochores were included in this dataset.

The anaphase speeds were measured on cells without merotelic kinetochores. Importantly, in the absence of a monastrol-release, the frequency of merotelic kinetochores is very low in RPE1 cells, and therefore less than 1% of the cells would be expected to have merotelic kinetochores.

8. The last sentence in the intro mentions “... for the first time ...”. This is not needed and likely not completely true. In this regard, the authors should refer to the MUG O’Connell/Khodjakov JCB 2008 paper somewhere in their work

We modified the last sentence accordingly, and now mention the MUG experiments in the introduction.

Reviewer #1 (Remarks to the Author):

I appreciate the effort put in by the authors to thoroughly address my (and the other reviewers) concerns. This resubmitted manuscript, with a significant amount of new and clearly presented data, is strengthened and adequately addresses my original concerns. I support publication of the work in Nature Communications and have only minor comments for the authors.

1) I still don't entirely buy the anaphase motility interpretation and do not think the calculation is quite as simple as inferring that MT number must account for 1.1 $\mu\text{m}/\text{min}$ since flux rates are reduced by 0.4 $\mu\text{m}/\text{min}$ (overall BAL reduction is 1.5 $\mu\text{m}/\text{min}$). The authors are welcome to make this inference in the discussion, but the addition of some caveats is warranted.

2) The statement on p.15, lines 383-384 page: "Our data further directly show that inter-kinetochore stretching is not required for rapid SAC satisfaction of wild-type kinetochores" cites the non-phosphorylatable Ndc80 mutant studies, but this was also a major conclusion of both the Uchida et al. and Maresca and Salmon JCB papers (2009) on kinetochore stretching/intra-kinetochore stretch.

Reviewer #2 (Remarks to the Author):

The authors have adequately addressed my concerns.

However, while reviewing the revised manuscript, I noticed a few minor issues (listed below) the authors may need to address prior to publication:

1. In figure 2i, the authors report new data on the velocity of chromosomes during metaphase oscillation. However, I was unable to find information about how this was quantified.

2. I noticed a formatting issue in the legend of figure 6.

3. I find that the very first sentence of the discussion is a little unclear. It would be beneficial if the authors could revise that sentence with a general (rather than expert) readership in mind.

Reviewer #3 (Remarks to the Author):

Overall the manuscript is much improved and we support its publication. We recommend the following minor changes:

1) We had previously written: "Finally, the authors need to write the anaphase section more clearly. Their current arguments assume that force and velocity are proportional, and I do not believe that a force-velocity relationship is known (e.g. it is possible that velocity does not change with force in a given regime)."

The authors replied: "Finally, on the relationship between force and velocity we respectfully disagree. As highlighted by the excellent review on anaphase forces of the Scholey laboratory (Civelekoglu-Scholey and Scholey, 2010 CMLS) forces and velocities are known to be proportional at the sub-cellular level: "In the sub-cellular regime, low Reynolds number conditions dominate motility, so that inertial forces (due to mass) are negligible in driving movements, and viscous forces (due to friction) have the major effect....This low value for the Reynolds number means that the chromosomes' movement is dominated by viscous forces, and hence the net force driving the movement of the chromosome is proportional to its velocity and its viscous drag coefficient: $F = \eta v$ ".

As discussed by Nicklas and others, the kinetochore may have a velocity-limiting governor such that more force on a kinetochore does not directly translate into a proportional change in velocity (e.g. see Nicklas, JCB 1983; Nicklas, Ann. Rev. Biophys. Biophys. Chem. 1988). We recommend that the authors remove their discussion addition that "Given that at the sub-cellular levels viscous forces dominate, and the speed of chromosome movements are directly proportional to the forces acting on them (for review see 73) we infer a 40% reduction in the forces pulling on chromosomes". Further, the authors should state that this inference ("we infer a 40% reduction in the forces pulling on chromosomes") assumes a proportional relationship between force and velocity – without saying that this assumption is correct (which is not known).

2) Figure 1 C-E: It would help to indicate where the quantified k-fiber intensity comes from for clarity (it is in the methods but it would also help to see this directly on the figure).

3) Figure 5 D and E: Although we agree that it seems challenging to unequivocally score merotelic attachments, it would help to understand roughly what fraction of the lagging chromosomes in the control monastrol-release condition result from merotelic-like attachment states. This is important to interpret whether the increase in lagging chromosomes in the BAL treated condition actually occurs from a higher frequency of merotelically or not. This seems like a core point of the paper and would benefit from more analysis (even if crude).

4) Figure S2: Missing inter-kinetochore distance labels on some panels of A.

Point by point response

We addressed the reviewers' final comments in the following manner:

Reviewer #1

I appreciate the effort put in by the authors to thoroughly address my (and the other reviewers) concerns. This resubmitted manuscript, with a significant amount of new and clearly presented data, is strengthened and adequately addresses my original concerns. I support publication of the work in Nature Communications and have only minor comments for the authors.

We appreciate that our new data helped to convince reviewer 1 that our work should be published in *Nature Communications*. We addressed the final minor comments in the following manner:

1) I still don't entirely buy the anaphase motility interpretation and do not think the calculation is quite as simple as inferring that MT number must account for 1.1 $\mu\text{m}/\text{min}$ since flux rates are reduced by 0.4 $\mu\text{m}/\text{min}$ (overall BAL reduction is 1.5 $\mu\text{m}/\text{min}$). The authors are welcome to make this inference in the discussion, but the addition of some caveats is warranted.

We agree that the calculation is not quite as simple and that the 1.1 $\mu\text{m}/\text{min}$ number might be misleading. Hence, we removed this number, and we now rather emphasize that the decrease in flux is relatively small compared to the decrease in anaphase A speed, and since a complete disruption of flux has been shown to only lead to a minor (20%) reduction in anaphase speed (Ganem et al., *Current Biol.* 2005), we conclude that the reduction in microtubule occupancy plays a primordial role in reducing anaphase A speeds.

2) The statement on p.15, lines 383-384 page: "Our data further directly show that inter-kinetochore stretching is not required for rapid SAC satisfaction of wild-type kinetochores" cites the non-phosphorylatable Ndc80 mutant studies, but this was also a major conclusion of both the Uchida et al. and Maresca and Salmon JCB papers (2009) on kinetochore stretching/intra-kinetochore stretch.

With regard to the Maresca and Salmon paper in JCB 2009, we fully agree, which is why we had already cited and mentioned this paper in the next sentence:

This is consistent with a study in Drosophila cells that found no correlation between inter-KT distances and SAC satisfaction³¹ (Reference 31 being the Maresca and Salmon paper).

We now have extended this sentence reporting the results from Uchida et al.

Reviewer #2:

The authors have adequately addressed my concerns. However, while reviewing the revised manuscript, I noticed a few minor issues (listed below) the authors may need to address prior to publication:

We are happy that reviewer 2 is satisfied with our revised manuscript. Below we addressed his/her last comments.

1. In figure 2i, the authors report new data on the velocity of chromosomes during

metaphase oscillation. However, I was unable to find information about how this was quantified.

This data was generated using our semi-automated kinetochore tracking code (Jaqaman et al., 2010 JCB). As we now briefly explain in the material and methods section, the code calculates the displacement along the spindle axis for each sister-kinetochore pairs, using the middle of the metaphase plate as the reference point. By plotting the distribution of all the displacements, one obtains a normal distribution; the standard deviation of this distribution is representative of the average speed at which sister-kinetochore move along the spindle axis.

2. I noticed a formatting issue in the legend of figure 6.

We thank the reviewer for spotting the formatting issue. It most likely comes from the conversion to PDF format. We will provide the figures directly in the “ai” format to avoid this issue.

3. I find that the very first sentence of the discussion is a little unclear. It would be beneficial if the authors could revise that sentence with a general (rather than expert) readership in mind.

We simplified the first sentence of the discussion as follows: “Our data indicate that the segregation of merotelically-attached kinetochores is favoured by high microtubule occupancy, which ensures a strong anaphase A force.” We thank the reviewer for making our paper more easily accessible to the broader readership.

Reviewer #3 (Remarks to the Author):

Overall the manuscript is much improved and we support its publication. We recommend the following minor changes:

We would like to thank the reviewer 3 for supporting the publication of our article in *Nature Communications*. We addressed the comments in the following manner.

1) We had previously written: “Finally, the authors need to write the anaphase section more clearly. Their current arguments assume that force and velocity are proportional, and I do not believe that a force-velocity relationship is known (e.g. it is possible that velocity does not change with force in a given regime).”

The authors replied: “Finally, on the relationship between force and velocity we respectfully disagree. As highlighted by the excellent review on anaphase forces of the Scholey laboratory (Civelekoglu-Scholey and Scholey, 2010 CMLS) forces and velocities are known to be proportional at the sub-cellular level: “In the sub-cellular regime, low Reynolds number conditions dominate motility, so that inertial forces (due to mass) are negligible in driving movements, and viscous forces (due to friction) have the major effect.... This low value for the Reynolds number means that the chromosomes’ movement is dominated by viscous forces, and hence the net force driving the movement of the chromosome is proportional to its velocity and its viscous drag coefficient: $F = lv$ ”.

As discussed by Nicklas and others, the kinetochore may have a velocity-limiting governor such that more force on a kinetochore does not directly translate into a proportional change in velocity (e.g. see Nicklas, JCB 1983; Nicklas, Ann. Rev. Biophys. Biophys. Chem. 1988). We recommend that the authors remove their discussion addition that “Given that at the sub-cellular levels viscous forces dominate, and the speed of chromosome movements are directly proportional to the forces acting on them (for review see 73) we

infer a 40% reduction in the forces pulling on chromosomes”. Further, the authors should state that this inference (“we infer a 40% reduction in the forces pulling on chromosomes”) assumes a proportional relationship between force and velocity – without saying that this assumption is correct (which is not known).

As suggested by the reviewer we now changed the mentioned sentence to:

“Assuming that at the sub-cellular levels viscous forces dominate, and that the velocity of chromosome movements are directly proportional to the forces acting on them (for review see 73) we infer a 40% reduction in the forces pulling on chromosome.”

This way we believe that we explain the logic of our conclusion, which is important for the general readership, while stating the linear relationship between force and velocity is an assumption.

2) Figure 1 C-E: It would help to indicate where the quantified k-fiber intensity comes from for clarity (it is in the methods but it would also help to see this directly on the figure).

We now put additional information in the figure legend about the way our code calculates the k-fiber intensities: (...) k-fibres were identified based on the CENPA signal and **the intensity of their kinetochore-proximal part (400nm from the CENPA centroid signal)** quantified with a custom written MATLAB-based code (>1000 kinetochore fibres per condition).

3) Figure 5 D and E: Although we agree that it seems challenging to unequivocally score merotelic attachments, it would help to understand roughly what fraction of the lagging chromosomes in the control monastrol-release condition result from merotelic-like attachment states. This is important to interpret whether the increase in lagging chromosomes in the BAL treated condition actually occurs from a higher frequency of merotelic or not. This seems like a core point of the paper and would benefit from more analysis (even if crude).

We agree with the reviewer that these points are worth clarifying. Lagging chromosomes have 3 possible sources: merotelically-attached chromosomes, non-disjoined chromosomes and acentric chromosome fragment that can arise due to chromosome breakages (Baudoin and Cimini, Chromosoma, 2018). Acentric chromosomes can be distinguished by the fact that they do not carry any centromere/kinetochore. From this current study and a previous study (Gasic et al., eLife, 2015), we know that nearly all lagging chromosomes contain a centromere/kinetochore; therefore acentric chromosomes can be neglected. We also know from Gasic et al. that in control monastrol-treated cells approximately 10% of the lagging chromosomes (2-3% of total cells) represent non-disjoined chromosomes, while 90% (or 20% of total cells) are true merotelic chromatids. Importantly, non-disjoined chromosomes are visible as two bulks of DNA in anaphase when staining with SIR-Hoechst. As we now indicate in the results, when quantifying lagging chromosomes in control- or BAL7826-treated cells, we did not count the rare lagging chromosomes showing two bulks of DNA (2-3% in either case). Rather we focused on the remaining lagging chromosomes, showing one DNA mass, which we assume are merotelic chromosomes. This assumption is validated by our immunofluorescence analysis, showing that the vast majority of lagging chromosomes that could be unambiguously classified, contained merotelically attached chromatids. We conclude that the BAL27862-treatment truly increases the absolute percentage of merotelic chromosomes, and that merotelic chromosomes represent the vast majority of lagging chromosomes in both control- and BAL27862-treated cells.

This clarification has now been also added to our manuscript on pages 11-12.

4) Figure S2: Missing inter-kinetochore distance labels on some panels of A.

We thank the reviewer for spotting this. We put back the labels.